# Insights into Some Onygenalean Fungi from Freshwater Sediments in Spain and Description of Novel Taxa

**DOI:** 10.3390/jof9121129

**Published:** 2023-11-22

**Authors:** Daniel Torres-Garcia, Josepa Gené, Dania García, Jose F. Cano-Lira

**Affiliations:** Unitat de Micologia i Microbiologia Ambiental, Facultat de Medicina i Ciències de la Salut and IU-RESCAT, Universitat Rovira i Virgili, 43201 Reus, Spain; daniel.torres@urv.cat (D.T.-G.); dania.garcias@urv.cat (D.G.); jose.cano@urv.cat (J.F.C.-L.)

**Keywords:** cycloheximide-resistant fungi, fluvial sediments, fungal diversity, *Onygenales*, phylogeny, taxonomy

## Abstract

During the course of a project investigating culturable *Ascomycota* diversity from freshwater sediments in Spain, we isolated 63 strains of cycloheximide-resistant fungi belonging to the order *Onygenales*. These well-known ascomycetes, able to infect both humans and animals, are commonly found in terrestrial habitats, colonizing keratin-rich soils or dung. Little is known about their diversity in aquatic environments. Combining morphological features and sequence analyses of the ITS and LSU regions of the nrDNA, we identified 14 species distributed in the genera *Aphanoascus*, *Arachniotus*, *Arthroderma*, *Arthropsis*, *Emmonsiellopsis*, *Gymnoascoideus, Leucothecium, Malbranchea*, and *Myriodontium*. Furthermore, three novel species for the genus *Malbranchea* are proposed as *M. echinulata* sp. nov., *M. irregularis* sp. nov., and *M. sinuata* sp. nov. The new genera *Albidomyces* and *Neoarthropsis* are introduced based on *Arachniotus albicans* and *Arthropsis hispanica*, respectively. *Neoarthropsis sexualis* sp. nov. is characterized and differentiated morphologically from its counterpart by the production of a sexual morph. The novel family *Neoarthropsidaceae* is proposed for the genera *Albidomyes*, *Apinisia*, *Arachnotheca*, *Myriodontium,* and *Neoarthropsis*, based on their phylogenetic relationships and phenotypic and ecological traits. *Pseudoamaurascopsis* gen. nov. is introduced to accommodate *P. spiralis* sp. nov., a fungus with unclear taxonomy related to *Amaurascopsis* and *Polytolypa*. We traced the ecology and global distribution of the novel fungi through ITS environmental sequences deposited in the GlobalFungi database. Studying the fungal diversity from freshwater sediments not only contributes to filling gaps in the relationships and taxonomy of the *Ascomycota* but also gives us insights into the fungal community that might represent a putative risk to the health of animals and humans inhabiting or transient in aquatic environments.

## 1. Introduction

Interest in fungi from aquatic environments has been increasing in recent years, although their diversity and functional roles are still poorly understood [1,2]. Recent evidence indicates that fungal diversity, particularly in freshwater environments, is high [2,3,4]. In fact, fungi play a critical role in those environments, degrading organic materials (i.e., plant litter and animal tissues) and contributing to key elemental cycles by releasing CO_2_ into the atmosphere and inorganic nitrogen and phosphorous into the soil [1,4,5,6,7]. It is not surprising, then, to find in freshwater sediments members of the order *Onygenales*. They have the ability to adapt to a wide range of environments, including oligo- and eutrophic habitats, and are also specialized in the degradation of compounds like cellulose from plant material and, particularly, keratin from animal waste [8].

The order *Onygenales* comprises a diverse group of ascomycetous fungi that are commonly found in terrestrial habitats worldwide. They usually inhabit soil and other substrates like the dung of herbivorous and carnivorous animals, which contain different sources of keratin as the primary source of environmental carbon and nitrogen [9,10,11,12,13,14,15]. Some well-delineated groups or families in *Onygenales* have clear ecological preferences in their ability to cause cutaneous (*Arthrodermataceae*) and systemic (*Ajellomycetaceae*) diseases in mammals, or to live closely associated with bees (*Ascosphaeraceae*) [16,17,18]. However, their diversity in freshwater sediments is still poorly studied.

Several studies have contributed to investigating the diversity of onygenalean fungi in freshwater sediments from different areas of the world [19,20,21,22,23,24,25,26]. Studies by Ulfig et al. [27,28,29,30] and Vidal [31] explored the diversity of culturable keratinolytic and nonkeratinolytic species of *Onygenales* isolated using various culture-dependent techniques (i.e., hair-baiting and agar media supplemented with cycloheximide) from fluvial sediments collected at the mouth of various Catalonian rivers (Spain). The predominant species were *Aphanoascus* (*Ap.*) *fulvescens*, *Ap. reticulisporus*, *Arthroderma* (*Ar.*) *curreyi*, *Chrysosporium* (*C.*) *keratinophilum*, *C. pannicola*, *C. lobatum*, *C. tropicum*, *Gymnoascus* (*G.*) *littoralis*, *Myriodontium* (*My.*) *keratinophilum*, *Narasimhella* (*N.*) *marginospora*, and *N. hyalinospora*, among others. However, in addition to those fungi, a set of unidentified isolates, recovered also from fluvial sediments at that time, were proposed in subsequent studies as novel taxa. These were *Arthropsis* (*A.*) *hispanica* [32], *C. fluviale* [33], *C. minutisporum* [34], *C. pilosum* [35], *C. submersum* [34], and *C. undulatum* [36], and the two members of the genus *Emmonsiellopsis* (*E.*), *E. coralliformis* and *E. terrestris* [37]. All these findings and others recently published [38,39,40,41,42,43,44] indicate that freshwater sediments serve as a reservoir for a great diversity of filamentous ascomycetes, which warrant further investigation in order to address gaps in fungal phylogenetic relationships, such as has been done in the present study for some species of the order *Onygenales*.

Of note is that the identification of many fungi from the abovementioned reports was based only on macro- and microscopic morphology. However, with the incorporation of molecular data in taxonomic studies, the classification of the aforementioned species and, in general, of the *Onygenales* and related taxa has changed considerably in recent decades. For instance, based primarily on sequence analyses of some regions of the nuclear ribosomal operon (i.e., internal transcribed spacer regions–ITS, the large subunit–LSU, and/or the small subunit–SSU of the rDNA), *C. pilosum* was found to be related to the genus *Arachnomyces* and renamed *Arachnomyces pilosus*, the genus that belongs to the order *Arachnomycetales* [45,46]; *C. minutisporum* and *C. submersum* were recently transferred to the genus *Keratinophyton* [47], as well as *C. pannicola* and *G. litoralis* to the genus *Arthroderma* and *Gymnascella*, respectively [8]. Meanwhile, the cosmopolitan, soilborne fungus *C. tropicum* is currently considered conspecific with *Ap. verrucosus*, and *N. marginospora* with *N. poonensis* [8]. Nevertheless, the taxonomy of other species like *A. hispanica* or numerous *Amauroascus* and *Chrysosporium* species remains unresolved because there are few specimens for studying and due to the polyphyletic nature of those genera [8]. Since morphological features used to classify the *Onygenales* [9] are insufficient for delimiting their taxa, polyphasic taxonomic studies integrating morphology, physiological traits, ecology, and phylogenetic relationships based on ITS and LSU sequence data have been crucial for a fairly stable classification in this order [15,48,49,50,51] and other fungi in general. In this scenario, the most recent reevaluation of the *Onygenales* has been conducted by Kandemir et al. [8]. Based on an integrative taxonomy, including the genomic data of various gene markers (i.e., ITS/LSU rDNA and partial fragments of the protein-encoding genes β-tubulin–*tub*2, elongation factor 1-α–*tef*1, partial fragments of the RNA polymerase I and II largest subunit–*rpb*1, *rpb*2, and ribosomal protein 60S L10–L1-rp60s), the latter authors elucidated the taxonomy and gave nomenclatural stability to numerous species and genera in the *Onygenales*. Furthermore, the new families *Malbrancheaeceae* and *Neogymnomycetaceae* were added to the seven existing families in the order (*Ajellomycetaceae*, *Arthrodermataceae*, *Ascosphaeraceae*, *Eremascaceae*, *Gymnoascaceae*, *Onygenaceae*, and *Spiromastigaceae*).

In the course of a project focused on the study of the diversity of culturable filamentous ascomycetes from freshwater sediments collected in Spanish rivers and streams, we recovered several specimens belonging to *Onygenales* with the use of different culture media, including potato dextrose agar (PDA) supplemented with cycloheximide. The aim of the present work was to resolve the taxonomy of the mentioned isolates, taking into account the integrative taxonomy described in the recent revision of this group of *Ascomycota* [8]. We also aim to assess the global distribution and habitat affinity of the newly proposed species here through the GlobalFungi database (https://globalfungi.com, accessed on 19 November 2023), which provides information on fungal distribution and ecology from metagenomic studies published over the last decade [52].

## 2. Materials and Methods

### 2.1. Sampling and Isolates

Sediments were collected between September 2018 and March 2021. Fungi included in this study were recovered from 21 sampled points in rivers and streams that flow through natural and rural areas of various Spanish provinces (Baleares, Barcelona, Castellón, Girona, Huesca, Lleida, Navarra, and Madrid; Figure 1). While samples from the streams were collected randomly, river sediments were mainly collected at the upper and middle sections of the rivers (Table 1).

Samples were collected and processed in the laboratory according to the procedures previously published [27,44]. The culture media used for fungal isolation were dichloran rose-bengal chloramphenicol agar (DRBC; 2.5 g peptone, 5 g glucose, 0.5 g KH_2_PO_4_, 0.25 g MgSO_4_, 12.5 mg rose bengal, 100 mg chloramphenicol, 1 mg dichloran, 10 g agar, 500 mL distilled water), DRBC supplemented with 0.01 g/L of benomyl, and potato dextrose agar (PDA; Condalab, Madrid, Spain) supplemented with 2 g/L of cycloheximide. The cultures were incubated at room temperature (22–25 °C) and examined weekly with a stereomicroscope for up to 4–5 weeks. To achieve pure cultures, colonies were transferred with a sterile dissection needle from isolation plate cultures to Petri dishes containing PDA with chloramphenicol and incubated at 25 °C in darkness.

All onygenalean isolates obtained in the present study (Table 1) were deposited in the fungal culture collection at the Medicine Faculty in Reus (FMR, University Rovira i Virgili, Reus, Catalonia, Spain). Holotypes, which consisted of dry cultures, and ex-type cultures of the novel species proposed here were also deposited at the fungal collection of the Westerdijk Fungal Biodiversity Institute (CBS; Utrecht, The Netherlands). Nomenclatural novelties were registered, and their respective descriptions deposited in Mycobank (https://www.mycobank.org/ (accessed on 31 July 2023)).

### 2.2. Phenotypic Study

For the in vitro macroscopic characterization, the isolates were inoculated on PDA, potato carrot agar (PCA; potato 20 g, carrot 20 g, agar 13 g, distilled water 1 L), oatmeal agar (OA; oatmeal 30 g, agar 13 g, distilled water 1 L), and phytone yeast extract agar (PYE; Pronadisa, Madrid, Spain). Colony features were obtained at 14 days, incubated at 25 °C in the dark. Color notations in descriptions were from Kornerup and Wanscher [53].

The microscopic characterization of the isolates studied was made from cultures growing on OA at 14 days at 25 °C in darkness and mounted onto wet slides, using water and 60% lactic acid. Photomicrographs were obtained using a Zeiss Axio-Imager M1 light microscope (Zeiss, Oberkochen, Germany) with a DeltaPix Infinity × digital camera. Photoplates were assembled from separate photographs using PhotoShop CS6 (Adobe Systems, San Jose, CA, USA).

In addition, to assess the ability of the novel fungi to grow at different temperatures, they were cultured in duplicate on PDA and incubated in the dark at 5 to 40 °C, in 5 °C intervals, including at 37 °C. Colony diameter was recorded at 7 and 14 incubation days.

### 2.3. DNA Extraction, Sequencing, and Phylogenetic Analysis

Isolates were cultured on PDA for 7–14 days at 25 °C in darkness. The DNA was extracted through the modified protocol of Müller et al. [54]. The extracted DNA was quantified by using Nanodrop 2000 (Thermo Scientific, Madrid, Spain). The loci amplified were the ITS regions and the LSU D1/D2 domain of the nrDNA using the primer pairs ITS5/ITS4 [55] and LR0R/LR5 [56,57], respectively. In addition, to achieve an accurate identification of the *Malbranchea* isolates, regions of the *tub*2 and *rpb*2 genes were also amplified with the primer pairs Bt2a/Bt2b [58] and RPB2-5F2 [59]/fRPB2-7cR [60], respectively. Briefly, PCR conditions for ITS, LSU, and *tub*2 were set as follows: an initial denaturation at 95 °C for 5 min, followed by 35 cycles of 30 s at 95 °C, 45 s at 56 °C, and 1 min at 72 °C, and a final extension step at 72 °C for 10 min, while the conditions for the *rpb*2 were an initial denaturation step at 94 °C for 5 min, followed by 5 cycles of 45 s at 94 °C, 45 s at 60 °C, and 2 min at 72 °C, then 5 cycles of 45 s at 94 °C, 45 s with 58 °C, and 5 min at 72 °C, followed by 30 cycles of 45 s at 95 °C, 45 s with 54 °C, and 2 min at 72 °C, and a final extension step at 72 °C for 7 min. Amplified products were purified and sequenced at Macrogen (Madrid, Spain). Consensus sequences were obtained using SeqMan v. 7.0.0 (DNAStar Lasergene, Madison, WI, USA).

In addition to a preliminary morphological identification, ITS sequences of the isolates under study were compared with those deposited in the National Center for Biotechnology Information (NCBI) using the Basic Local Alignment Search Tool (BLAST; https://blast.ncbi.nlm.nih.gov/blast.cgi (accessed on 28 December 2022)). A maximum similarity level of >98% (≥90 sequence coverage) was used for species-level identification, while lower similarity values were considered to be sequences belonging to putative novel fungi. To assess taxonomic positions and delineate novel taxa within *Onygenales*, we carried out two different phylogenetic analyses. The first was constructed using the ITS and LSU phylogenetic markers and included species representatives of the different families and monophyletic clades recognized in Kandemir et al. [8] (i.e., *Ajellomycetaceae, Arthrodermataceae, Ascosphaeraceae, Eremascaceae, Gymnoascaceae, Malbrancheaceae, Neogymnomycetaceae, Onygenaceae, Spiromastigoidaceae*, and related *incertae sedis* clades). A second phylogenetic reconstruction was carried out using the ITS, LSU, *tub*2, and *rpb*2 markers for members of *Malbrancheaceae*. Sequences of the ex-type strains and/or reference strains of accepted species and genera of the aforementioned families used in the analyses, including taxa selected as outgroups, were retrieved from GenBank (Appendix A).

Datasets for each locus were aligned individually in MEGA (Molecular Evolutionary Genetics Analysis) software v.6.0 [61], using the ClustalW algorithm [62] and refined with MUSCLE [63] or manually adjusted, if necessary, on the same platform. Before combining the regions, the phylogenetic concordance of the loci datasets was assessed individually for each single-locus phylogeny through visual comparison and using the Incongruence Length Difference (ILD) run in Winclada v.1.00.08 software [64] in order to determine any incongruent results among nodes with high statistical support. Once a lack of incongruence was confirmed, individual alignments were concatenated into a single data matrix with SequenceMatrix v.1.7.6 software [65]. The best substitution model for all gene matrices was estimated using MEGA software for maximum likelihood (ML) analysis, whereas for the Bayesian Inference (BI) analysis, it was estimated using jModelTest v.2.1.3 according to the Akaike criterion [66]. The phylogenetic reconstructions were made with the combined genes using ML under RaxML-HPC2 on XSEDE v-8.2.12 [67] in the CIPRES Science gateway portal [68] and BI with MrBayes v.3.2.6 [69]. The phylogenetic support of internal branches for ML analyses was assessed by 1000 ML bootstrapped pseudoreplicates and a bootstrap support (bs) of ≥70 was considered significant [70]. The phylogenetic reconstruction by BI was carried out using 5 million Markov chain Monte Carlo (MCMC) generations, with four runs (one cold chain and three heated chains), and samples were stored every 1000 generations. The 50% majority-rule consensus tree and posterior probability (pp) values were calculated after discarding the first 25% of samples. A pp value of ≥0.95 was considered significant [71]. The resulting trees were plotted using FigTree v.1.3.1 (http://tree.bio.ed.ac.uk/software/figtree/ (accessed on 15 February 2023)).

DNA sequences and alignments generated in this study were deposited in GenBank (Table 1) and in Zenodo under the submission number 10036241 (https://doi.org/10.5281/zenodo.10036241), respectively.

### 2.4. Phylogeny and Geographical Distribution of the Related Environmental Sequences

To assess the global geographical distribution and habitat affiliations of the novel taxa among environmental sequences, we followed the workflow of Réblová et al. [72,73] and Torres-Garcia et al. [42]. The full length of the ITS1 and ITS2 sequences of our isolates were blasted against those deposited in the GlobalFungi database (https://globalfungi.com (accessed on 2 April 2023)) [52]. At the time of the approach, this database contained 57,184 samples from 515 studies, 791,513,743 unique sequences for ITS1, and 2,892,377,338 for ITS2. As this database has separated ITS1 and ITS2 sequences, they were processed separately. In order to verify the generic and species boundaries among the downloaded ITS environmental sequences related to our fungi, we also included in the analyses ITS sequences of known species previously obtained from GenBank. Those known species were representatives of the well-delineated closest taxa to our fungi in a full ITS analysis carried out previously (Appendix A). The ITS1 and ITS2 sequences of some of those fungi were also blasted against the GlobalFungi database and included in the respective analyses. For both markers, ITS1 and ITS2, we selected and downloaded environmental sequences that had a similarity between 98 and 100% and a full-length coverage with those of the sediment isolates and related known species. Due to the large numbers of environmental sequences obtained, we selected those representative of various geographical origins, biomes, and substrates. For each taxon, we obtained data on their occurrence across environmental samples. Metadata related to the particular samples (location, climatic data, biome, or substrate) is listed in Appendix A).

## 3. Results

After a preliminary morphological examination of the primary cultures, we recovered 63 isolates of onygenalean fungi from freshwater sediments (57 on PDA supplemented with 0.2% cycloheximide, four on DRBC, and two on DRBC supplemented with benomyl), of which 55 were identified at the species level through ITS sequence comparison. These isolates were representatives of 14 species of *Onygenales*, i.e., *A. hispanica* (n = 3), *Ap. crassitunicatus* (n = 3), *Ap. fulvescens* (n = 11), *Ap. reticulisporus* (n = 4), *Ar. curreyi* (n = 1), *Arachniotus* (*Ara.*) *albicans* (n = 12), *E. tuberculata* (n = 1), *Gymnoascoideus* (*Gym.*) *petalosporus* (n = 1), *Leucothecium* (*L.*) *emdenii* (n = 1), *M. chinensis* (n = 1), *M. ostraviensis* (n = 1), *M. reticulata* (n = 1), *M. umbrina* (n = 3), and *My. keratinophilum* (n = 12). However, eight of the isolates could not be assigned to any known species (ITS similarity values ≤ 98%) despite exhibiting a malbranchea- (FMR 17906, FMR 18266, FMR 19015, FMR 19016, FMR 19017, FMR 19030), arthropsis- (FMR 19025), and amaurascopsis-like (FMR 19014) morphology.

Table 1 includes the final identification of the isolates that resulted from the phenotypic and phylogenetic characterization.

### 3.1. Phylogeny

ITS and LSU sequences of the sediment isolates, apart from those considered isogenic, were analyzed phylogenetically in order to confirm identification and assess the taxonomic position of the unidentified isolates. In addition, since *tub*2 and *rpb*2 sequences were available for the currently accepted *Malbranchea* species (Appendix A), we also carried out a multi-gene analysis combining the ITS, LSU, *tub*2, and *rpb*2 loci for a more precise delineation of the unidentified malbranchea-like isolates. The tree topology was in all cases similar for the ML and BI methods, so the ML trees were selected to represent our results. Bootstrap and BI posterior probability values are shown on the relevant branches.

The ITS/LSU phylogenetic tree of the order *Onygenales* is shown in Figure 2. It encompasses 162 isolates, representatives of most genera accepted in the order, with 318 sequences that comprised 1118 bp (520 for ITS and 598 for LSU), of which 699 (440 for ITS and 259 for LSU) corresponded to variable sites, and 589 (385 for ITS and 204 for LSU) were phylogenetically informative sites. For ML analysis, TN93+G+I was selected as the best substitution model, while for the BI analysis, GTR+G+I was selected as the best-fit model for ITS and GTR+G for LSU. This phylogeny agrees with the multi-locus analysis of the *Onygenales* published by Kandemir et al. [8]. There are similar topologies regarding the monophyletic clades representative of currently accepted families in the order, as well as other well-supported clades that represent putative undescribed families (Clades I–XIII). The only exceptions were the unresolved taxonomic Clades VI and VII, with the genera *Leucothecium* and *Myotisia* in the first and *Shanorella* in the latter, which seem to be members of the family *Arthrodermataceae* (Clade V).

Sediment isolates were distributed in the following clades: Clade I (*Malbrancheaceae*), Clade III (*Onygenaceae*), Clade V (*Arthrodermataceae*), Clade VIII (*Gymnoascaceae*), Clade IX (*Ajellomycetaceae*), and Clades IV, VI, X (*incertae sedis*) according to Kandemir et al. [8].

**Figure 2 jof-09-01129-f002:**
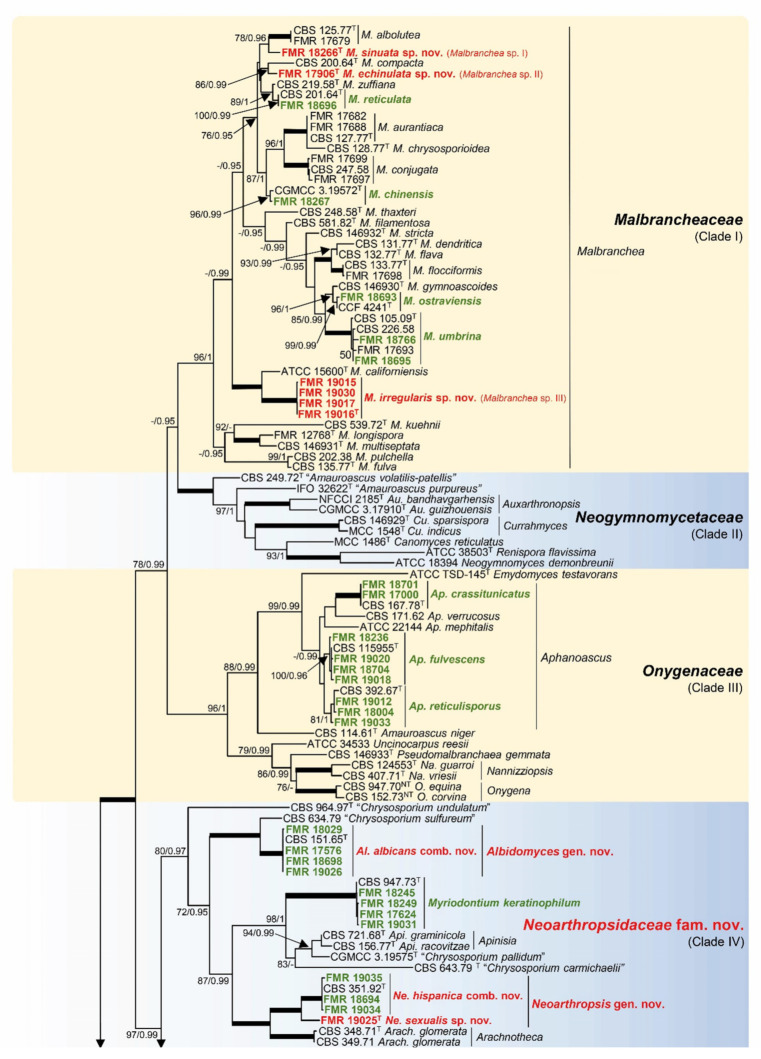
Phylogenetic tree based on maximum likelihood (ML) analysis obtained by RAxML inferred from combined ITS and LSU sequences of the *Onygenales* and outgroups. Branch lengths are proportional to phylogenetic distance. Bootstrap support values/Bayesian posterior probability scores above 70%/0.95 are indicated on the nodes. Bold branches indicate bs/pp values 100/1. The tree was rooted to several members of the orders *Arachnomycetales* and *Eurotiales*. Quote marks indicate species with unresolved taxonomy. The novel taxa proposed are in red, and the known species identified in this study are in green. ^T^= ex−type strain.

Among malbranchea-like isolates included in the *Malbrancheaceae*, four species were identified: *M. chinensis* (FMR 18267), *M. ostraviensis* (FMR 18693), *M. reticulata* (FMR 18696), and *M. umbrina* (FMR 17899, FMR 18695, and FMR 18766). Although the isolates FMR 17906, FMR 18266, FMR 19015, FMR 19016, FMR 19017, and FMR 19030 were confirmed as members of the genus *Malbranchea*, they were distributed in three distinct branches representative of putative undescribed *Malbranchea* species (I–III). These novelties were confirmed with phylogeny based on the ITS, LSU, *rpb*2, and *tub*2 genes, in which the accepted species of *Malbranchea* were compared (Figure 3). This phylotree encompasses 50 strains, including the selected outgroups. It includes 139 sequences that comprised 2749 bp (547 for ITS, 511 for LSU, 1079 for *rpb*2, 612 for *tub*2), of which 1110 (287 for ITS, 130 for LSU, 366 for *rpb*2, 327 for *tub*2) corresponded to variable sites, and 793 (211 for ITS, 113 for LSU, 295 for *rpb*2, 174 for *tub*2) were phylogenetically informative sites. For ML analysis, K2+G+I was selected as the best substitution model, while for the BI analysis, GTR+G+I was selected as the best-fit model for ITS, SYM+G for LSU, and K2+G for *tub*2 and rpb2. Unfortunately, *rpb*2 and *tub*2 sequences for the species *M. albolutea*, *M. californiensis*, and *M. compacta* were not available for comparison. *Malbranchea* sp. I (FMR 18266) formed a singleton distant branch into a statistically supported terminal clade (89 bs/0.99 pp) together with *M. albolutea* (CBS 125.77 ex-type, FMR 17679). Both specimens showed enough distance (98.8–98.9% for LSU and 95.7–95.8% for ITS) to be considered distinct species. The phylogenetic distance and morphological differences observed with its counterpart (see taxonomy section) support the proposal of the new species *Malbranchea sinuata*. *Malbranchea* sp. II (FMR 17906) was part of a supported terminal clade (81 bs/0.99 pp), but forming a singleton lineage with significant genetic distance (99.4% for LSU and 96.2% for ITS) regarding the ex-type strain of *M. compacta* (CBS 200.64). It is therefore proposed here as *Malbranchea echinulata* sp. nov. *Malbranchea* sp. III (FMR 19015, FMR 19016, FMR 1907, and FMR 19030) formed a fully supported terminal clade (100 bs/1 pp) closely related to the ex-type strain of M*. californiensis* (ATCC 15600) but representing an undescribed lineage for the genus. The phylogenetic distance from its counterpart (98.9–99.0% for LSU and 90.5–90.6% for ITS) and different morphological features (see taxonomy section) allow us to propose *Malbranchea irregularis* sp. nov.

The well-supported clade (96% bs/1 pp) of the family *Onygenaceae* (Figure 2) included the sediment isolates that were identified morphologically as belonging to the genus *Aphanoascus*. The species confirmed molecularly through representative sediment isolates were *Ap. crassitunicatus* (FMR 18700 and FMR 18701), *Ap. fulvescens* (FMR 18236, FMR 18704, FMR 19018, and FMR 19020), and *Ap. reticulisporus* (FMR 18004, FMR 19012, and FMR 19033).

Clade IV (Figure 2; 80% bs/0.97 pp) comprised a rich set of specimens belonging to various taxa that were repeatedly isolated from freshwater sediments. We recovered numerous isolates of *A. hispanica*, *Ara*. *albicans*, and *My. keratinophilum* (Table 1). These species were confirmed molecularly through representative isolates included in the concatenated ITS/LSU analysis. It is noteworthy that the protologue of *A. hispanica* was described based on freshwater and marine sediments collected in Spain by Ulfig et al. [32]. Another fungus included in this clade and also described from aquatic sediments was *Chrysosporium* (*C.*) *undulatum* [36]. These data suggest that a good number of the members of this clade seem to be well adapted to aquatic environments. Nevertheless, the species of *Apinisia* or *Arachnotheca*, two genera also included in this *incertae sedis* clade, have been recovered from soil, plant debris, and animal skin [8,74].

Of note is that ex-type or reference strains of several species that require taxonomic adjustment are included in Clade IV, as already reported by Kandemir et al. [8]. Those include *A. hispanica*, *Ara. albicans*, and the previously mentioned *C. undulatum*, as well as *C. carmichaelii*, *C. pallidum,* and *C. sulfureum*; several of these species are placed in phylogenetically distant branches that might represent undescribed genera per se (Figure 2). We have studied, morphologically and molecularly, the former two species. The isolates identified as *A. hispanica* (18694, FMR 19034, and FMR 19035) clustered in a fully supported terminal clade with the ex-type strain of the species (CBS 351.92). This terminal clade together with the separate singleton branch of the unidentified arthropsis-like isolate (FMR 19025) formed a monophyletic lineage at a significant genetic distance (88.12–88.33% of similarity for ITS and 96.29–96.92% for LSU) from the lineage representative of the monotypic genus *Arachnotheca* [74]. The phylogenetic distance and morphological differences observed between the members of the two lineages (see taxonomy section) provide evidence for proposing the novel genus *Neoarthropsis* for the *Arthropsis* specimens since the generic type *A. truncata* is allocated in the *Sordariomycetes.* The genus is typified with *A. hispanica*, and the new combination *Neoarthropsis* (*Ne.*) *hispanica* is proposed. The phylogenetic distance between the ex-type of *Ne. hispanica* and the unidentified isolate FMR 19025, with similarity values of 91.2% for ITS and 98.5% for LSU, supports the proposal of the novel species *Neoarthropsis sexualis.*

**Figure 3 jof-09-01129-f003:**
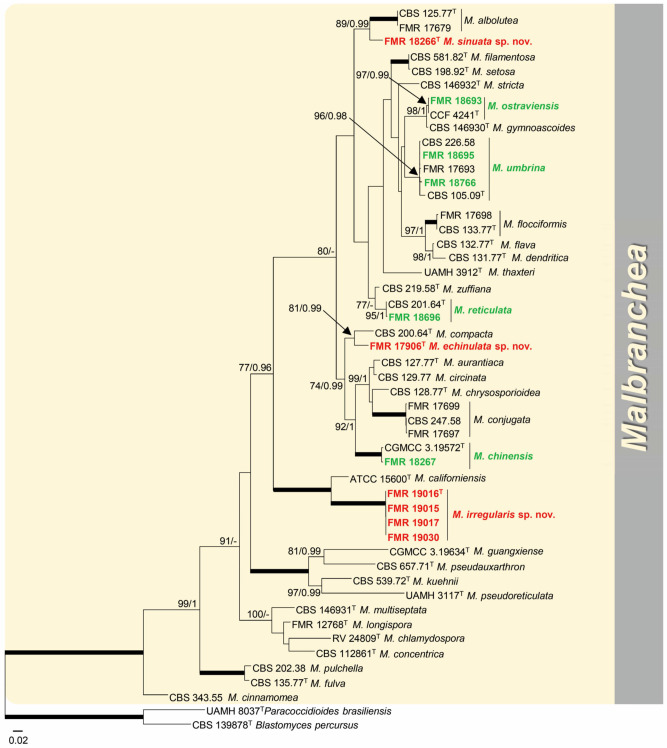
Phylogenetic tree based on a maximum likelihood (ML) analysis obtained by RAxML inferred from combined ITS, LSU, *tub*2, and *rpb*2 sequences of members of the genus *Malbranchea* and outgroups. Branch lengths are proportional to phylogenetic distance. Bootstrap support values/Bayesian posterior probability scores above 70%/0.95 are indicated on the nodes. Bold branches indicate bs/pp values 100/1. The tree was rooted to *Paracoccidioides brasiliensis* UAMH 8037 and *Blastomyces percursus* CBS 139878. The novel taxa proposed are in red, and the known species identified in this study are in green. ^T^ = ex−type strain.

Representative sediment isolates of *Ara. albicans* (FMR 17576, FMR 18029, FMR 18698, and FMR 19026) grouped with the ex-type strain of this species (CBS 151.65) to form, along with a reference strain of *C. sulfureum*, an undescribed lineage that can be distinguished easily from the others in Clade IV. Considering that the type species of the genus *Arachniotus* is placed in the family *Gymnoascaceae* [75] (Clade VIII), we proposed the new genus *Albidomyces*, typified by *Albidomyces* (*Al.*) *albicans* comb. nov. Since Clade IV comprises a highly supported group of taxa distantly related to other families in *Onygenales*, whose members share morphological (see taxonomy section) and ecological traits as mentioned above, it deserves to be recognized as a novel family, named *Neoarthropsidaceae*. Taxa delineation in this new family is also supported by an additional ITS analysis (Appendix A), in which more isolates from different origins were added for several species.

The ITS/LSU analysis also confirmed the identity and taxonomic position of *Ar. curreyi* (FMR 19038) in the *Arthrodermataceae* (Clade V), *L. emdenii* (FMR 18703) in the *incertae sedis* Clade VI, *Gym. petalosporus* (FMR 19036) in the *Gymnoascaceae* (Clade VIII), and *E. tuberculata* (FMR 17582) in the *Ajellomycetaceae* (Clade IX). The asexual morph of *L. emdenii*, under the name *Arthrographis alba*, was repeatedly isolated from freshwater and marine sediments also collected in Spain [76].

The remaining unidentified isolate amaurascopsis-like strain (FMR 19014) clustered into the well-supported *incertae sedis* Clade X (Figure 2; 99% bs/1 pp). This clade included the ex-type strain of *Amaurascopsis* (*Am*.) *perforata* (FMR 3882), but also those of *Polytolypa* (*Po*.) *hystricis* (UAMH 7299) and *C. chiropterorum* (MUCL 45495), as well as a reference strain of *C. lobatum* (CBS 275.77). The closest species to the unidentified isolate were *Am. perforata* and *Po. hystricis*, but both fungi showed a sequence similarity of 89.2% for ITS and of 97.4% for LSU in the case of *Po. hystricis*. The LSU sequence of *Am. perforata* was not available for comparison. The phylogenetic distance as well as morphological differences, mainly regarding growth temperatures and conidial morphology (see taxonomy section), support the proposal of the new genus *Pseudoamaurascopsis* to accommodate *Pseudoamaurascopsis* (*Ps.*) *spiralis* sp. nov. *Chrysosporium* species included in the clade deserve further studies to clarify their taxonomy. Although the phylogeny of Clade X suggests that it could represent an undescribed family, the small number of taxa and the long phylogenetic distance between them preclude a formal taxonomic proposal for the time being.

### 3.2. Distribution of Species in the Rivers and Streams Sampled

The isolates under study were mainly recovered from samples collected in the rivers (n = 50; 79.4%) as they pass through various Spanish provinces, i.e., Llobregat (Barcelona) (n = 20; 31.8%), Segre (Lleida) (n = 17; 27.0%), and Ter (Barcelona and Girona) (n = 13; 20.6%). Meanwhile, the isolates obtained from streams (n = 13; 20.6%) were distributed as follows: Torrent de la Coma (Lleida; n = 4, 6.3%), Mare de Déu Stream (Castellón; n = 3, 4.7%), Remáscaro Stream (Huesca; n = 2, 3.2%), Brezal Stream (Madrid; n = 1, 1.6%), Esca Stream (Navarra, n = 1, 1.6%), Pont d’en Blai Stream (Mallorca; n = 1, 1.6%), and Umbría Stream (Madrid; n = 1, 1.6%). The difference in the number of isolates between rivers and streams is probably influenced by the greater number of sampling points in the rivers.

Although the number of isolates was rather similar among the three rivers, samples from the rivers Llobregat and Ter exhibited the greatest number of species identified, i.e., 10 species in the Llobregat (i.e., *Al. albicans*, *Ap. fulvescens*, *Ap. reticulisporus*, *Ar. curreyi*, *Gym. petalosporus*, *M. irregularis*, *My. keratinophilum*, *Ne. hispanica, Ne. sexualis*, and *Ps. spiralis*) and eight in the Ter (i.e., *Al. albicans*, *Ap. crassitunicatus*, *Ap. fulvescens*, *L. emdenii*, *M*. *ostraviensis*, *M. reticulata*, *M. umbrina*, and *Ne. hispanica*). Meanwhile, only four species were recovered in the river Segre (i.e., *Ap. fulvescens*, *M. chinensis*, *M. sinuata*, and *My*. *keratinophilum*). Curiously, five out of the seven novel taxa proposed here (i.e., Al. albicans, *M. irregularis*, *Ne. hispanica*, *Ne. sexualis*, and *Ps. spiralis*) were isolated from samples collected in the river Llobregat at the Balsareny locality (Catalonia).

As mentioned previously, *Al. albicans*, *Ap. fulvescens*, and *My. keratinophilum* were the most common species isolated, with 11 or 12 isolates for each species. However, their distribution across the sampling points showed some differences. While *Al. albicans* was found in 7 out of the 21 sampled points from different rivers and streams, and mainly recovered along the rivers Llobregat and Ter, *Ap. fulvescens* and *My. keratinophilum* were found in 6 and 5 sampled points, respectively, but both species were primarily recovered from samples collected along the river Segre, in particular the latter species (Table 1).

### 3.3. Global Biogeography and Ecology of the Novel Taxa

According to BLAST results with ITS sequences in the GlobalFungi database, we were able to plot the global distribution of the following novel taxa through environmental sequences of samples collected around the world: *Al. albicans*, *M. sinuata*, *Ne. hispanica*, *Ne. sexualis*, and *Ps. spiralis* (Figure 4). However, hindered environmental sequences of undescribed hypothetical species related to the aforementioned taxa were not detected in our ITS 1 and ITS 2 analyses (Appendix A).

ITS1 sequences of *Al. albicans*, *Ne. sexualis*, and *Ps. spiralis* were recovered from 173, 84, and 270 environmental samples, respectively (similarity 89–100%). The high amounts of environmental sequences obtained after BLAST regarding our studied fungi allowed us to select the sequences that originated from various environmental samples. Additionally, other environmental sequences of the closest species were added to the reconstructed phylogeny (Appendix A). ITS1 sequences of these three species were retrieved from soil samples collected from biomes in various European countries: in particular, those linked to *Al. albicans* were from grassland, forest, and urban environments in Estonia, the Czech Republic, and Latvia; those linked to *Ne*. *sexualis* were retrieved from grassland soils in Estonia; and those of Ps. Spiralis were found in samples collected from soil, roots, and rhizosphere soil of croplands, grasslands, and urban environments in Denmark, Estonia, Spain, and the UK. In addition, environmental sequences of the hypothetical species *Ne. sexualis* were also detected in soil samples from Australia, and of *Ps. spiralis*, in soil from Australia and China.

On the other hand, the ITS2 sequences give us additional biogeography and ecological information for those three species, which were recovered from 478, 20, and 632 environmental samples, respectively (similarity 98–100%). Environmental ITS2 sequences of *Al. albicans* were obtained from samples of soil and rhizosphere soil of different environments (grassland, forest, and woodland) exclusively in Europe (the Czech Republic, Estonia, Germany, and the UK). Meanwhile, environmental ITS2 sequences linked to *Ne*. *sexualis* and *Ps. spiralis* revealed that both are soil-inhabiting fungi, but those of the former species were found mainly in samples of forest soil in the USA, while those of *Ps. spiralis* were associated with cropland soil samples from Australia, China, and the USA (Figure 4).

The ITS sequences of *M. echinulata* and *M. irregularis* did not match any ITS1/ITS2 environmental sequences deposited in the GlobalFungi database. As mentioned previously, only environmental ITS2 sequences deposited in the database corresponded with the ITS2 sequence of *M. sinuata* (Appendix A), which were recovered from 66 environmental samples (98–100% similarity). Its geographical distribution was limited to samples from soil and rhizosphere soil of shrublands in Spain (Figure 4).

### 3.4. Taxonomy

This section is organized according to the phylogenetic position of the family shown in our concatenated ITS and LSU analysis, and in alphabetical order of the respective novel genera and species proposed.

***Malbrancheaceae*** (Figure 2, Clade I)***Malbranchea echinulata*** Torres-Garcia, Cano & Gené, sp. nov. Figure 5.

MycoBank MB848037

*Etymology*: Referring to the wall ornamentation of the peridial hyphae and ascospores.

*Type*: Spain, Comunidad de Madrid, Rascafría, Guadarrama Natural Park, Arroyo de la Umbría, N 40°51′54.7′′ W 3°53′40.3′′, from fluvial sediments, May 2019, J. Cano (holotype CBS H-25255; cultures ex-type FMR 17906, CBS 149936).

*Description: Mycelium* immersed and superficial, composed of hyaline, septate, branched, smooth-walled, 1.5–2.5 μm wide hyphae. *Asexual morph* with well-differentiated fertile hyphae arising laterally from vegetative hyphae, branched, sinuous and forming randomly intercalary and terminally arthroconidia, 1.5–2.5 μm wide. *Arthroconidia* enteroarthric, 0–1-septate, hyaline, smooth- and thin-walled, cylindrical, subcylindrical or T-shaped, 2.5–6 × 1–3 μm; secession rhexolytic. *Sexual morph* with gymnothecial ascomata, globose, single to confluent, brown, 182–432 μm diam. (excluding appendages); peridial hyphae branched, septate, pale orange-brown, finely echinulate, thick-walled, 2–3.5 μm wide, with short and long lateral appendages; short appendages blunt, spine-like, with both subacute or truncate ends; long appendages 2–5, arising from peridium at acute angles, unbranched, straight and with subacute ends, orange-brown, paler toward apex, smooth-walled, mostly with one basal knuckle-joint, 127–778 μm long. *Asci* 8-spored, evanescent, hyaline, irregularly disposed, globose to subglobose, 6–8.5 × 6–8 μm. *Ascospores* hyaline, coarsely echinulate, thick-walled, globose, 2.5–3 μm diam.

*Culture characteristics (14 d at 25 °C)*: Colonies on PDA reaching 43–44 mm diam., slightly elevated, dense, cottony, concentrically zonate, pastel yellow (1A4) at the center, yellowish white (1A2) to white toward the periphery, margins regular, sporulation abundant, reverse olive brown (4D4) at the center, yellowish white (5F6) toward the periphery, diffusible pigment absent. On PCA, reaching 47–48 mm diam., slightly elevated, floccose, brownish gray (8C2), white (1A1) at the periphery, margins regular, sporulation abundant, reverse brownish gray (8D2) at the center, yellowish white (1A2) to white (1A1) toward the periphery, diffusible pigment absent. On OA, reaching 29–30 mm diam., flattened, aerial mycelium sparse, granulose due to presence of ascomata, brown (7E5), whitish at the periphery, margins irregular, sporulation abundant, reverse grayish orange (5B4) at the center, diffusible pigment absent. On PYE, reaching 20–21 mm diam., elevated, cottony, slightly concentrically sulcate, yellowish white (1A2) at the center, colorless toward the periphery, margins fimbriate, sporulation sparse to moderate; reverse yellow (5A6) at the center, light yellow (4A6) toward the periphery, diffusible pigment absent.

*Cardinal temperatures for growth on PDA after 14 d (mm)*: Minimum 15 °C (18–20), optimum 25 °C (42–43), maximum 30 °C (12–14).

*Habitat and geographical distribution*: Freshwater sediments, Spain.

*Notes: Malbranchea echinulata* is related to *M. compacta* (=*Auxarthron compactum*) but with a phylogenetic distance far enough to be considered a distinct species (Figure 3). Both species are similar in forming ascomata with pale orange-brown peridial hyphae that show a dense, net-like reticulum with short and long appendages. However, *M. echinulata* has smaller ascomata (182–432 μm diam vs. 310–675 μm diam. in *M. compacta*) and considerably longer (127–778 μm) and straight appendages, while those of *M. compacta* are up to 600 μm long and bent to coiled apically [74,77]. The ascospores of *M. compacta* are slightly larger (up to 3.6 μm diam), hyaline to pale brown, and delicately asperulate to echinulate [77], while those of *M. echinulata* are 2.5–3 μm, hyaline, and coarsely echinulate. It is difficult to establish differences between their asexual morphs because of the variable size and shape of the arthorconidia, although they are hyaline to pale yellow in *M*. *compacta* [77], while strictly hyaline in *M. echinulata*.

***Malbranchea irregularis*** Torres-Garcia, Gené & Dania García, sp. nov. Figure 6.

MycoBank MB848039

*Etymology*: Referring to the irregular shape of the colonies on PYE and PDA.

*Type*: Spain, Catalonia, Berguedà, Castellar de n’Hug, Llobregat River, N 42.27298° E 1.99177°, from fluvial sediments, March 2021, *D. Torres-Garcia & J. Gené*, (**holotype** CBS H-25256; cultures ex-type FMR 19016, CBS 149937).

*Description: Mycelium* immersed and superficial, composed of hyaline, septate, branched, smooth-walled, 1–2.5 μm wide hyphae. *Sexual morph* with gymnothecial ascomata, globose, 134–317 μm diam. (excluding appendages), rosaceous initially, becoming brown to orange-brown, peridial hyphae asperulate to verrucose, 3–5 μm wide, with short and long lateral appendages, short appendages orange-brown, verruculose, spine-like, with subacute to rounded ends, 23–29 μm long, long appendages arising from peridium at acute and subacute angles, unbranched, straight, cylindrical, progressively tapering terminally, orange-brown, thick-walled, and asperulate toward the base, paler and smooth to asperulate terminally, with a rounded or subacute apex and a basal knuckle-joint, 42.5–230.5 μm long. *Asci* 8-spored, evanescent, irregularly disposed, hyaline, globose to subglobose, 6–9 × 5.5–7.5 μm. *Ascospores* hyaline, minutely echinulate to echinulate, thick-walled, globose, 2.5–3 μm diam. *Asexual morph* not observed.

**Figure 5 jof-09-01129-f005:**
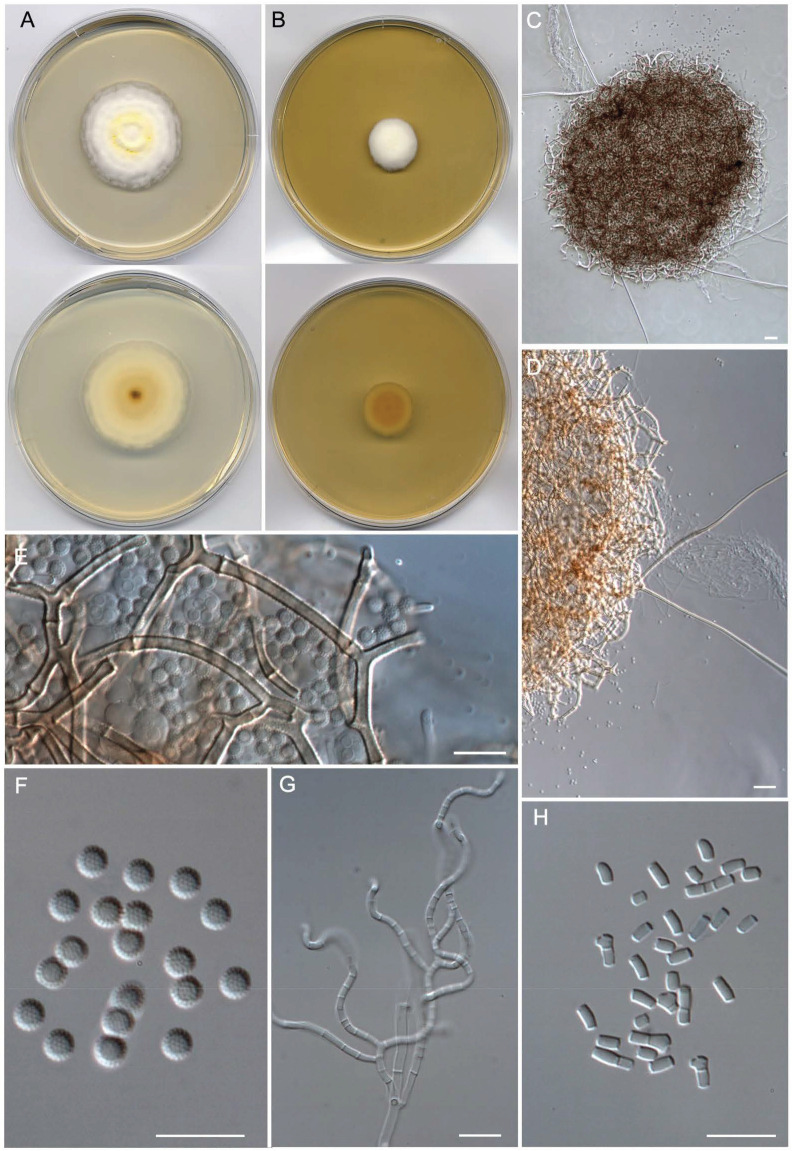
*Malbranchea echinulata* (ex-type FMR 17906). (**A**) Colonies on PDA (front and reverse) at 25 °C after 14 days. (**B**) Colonies on PYE (front and reverse) at 25 °C after 14 days. (**C**,**D**) Ascomata. (**E**) Peridial hyphae, asci, and ascospores. (**F**) Ascospores. (**G**) Fertile hyphae. (**H**) Arthroconidia. Scale Bars: (**C**) = 75 μm; (**D**) = 25 μm; (**E**−**H**) = 10 μm.

*Culture characteristics (14 d at 25 °C)*: Colonies on PDA reaching 21–22 mm diam., slightly raised at the center, velvety, yellowish white (3A2) to light yellow (3A4), margins irregularly lobulated, sporulation absent, reverse light brown (5D6) to golden yellow (5B7), diffusible pigment absent. On PCA, reaching 11–12 mm diam., slightly raised at the center, velvety, grayish orange (5B5) at the center, pale yellow (4A3) toward the periphery, margin lobulated, sporulation absent to sparse, reverse yellowish brown (5E6) at the center, orange (5A2) at the periphery, diffusible pigment absent. On OA, reaching 19–20 mm diam., flattened, granular, light brown (6D4) to brownish orange (6C4) at the center, yellowish white (1A2) at the periphery, margins fimbriate, sporulation moderate to abundant, reverse grayish yellow (4B4), diffusible pigment absent. On PYE, reaching 22–23 mm diam., irregularly elevated, radially sulcate at the periphery, velvety, grayish yellow (4B4) at the center, yellow (4A6) to pale yellow (4A3) toward the periphery, margins irregularly lobulated, sporulation absent, reverse dark brown (6F8) at the center, pale brown (5D7) toward the periphery, diffusible pigment absent.

*Cardinal temperatures for growth on PDA after 14 d (mm)*: Minimum 5 °C (2–3), optimum 25 °C (25–26), maximum 30 °C (6–7).

*Additional specimens examined*: Spain, Catalonia, Berguedà, Balsareny, Llobregat River, N 41.87818° E 1.88900°, from river sediments, March 2021, *D. Torres-Garcia & J. Gené* (FMR 19015); ibid. Castellar de n’Hug, Llobregat River, N 42.27967° E 2.00535°, from river sediments, March 2021, *D. Torres-Garcia & J. Gené* (FMR 19017); ibid. (FMR 19030).

*Habitat and geographical distribution*: Freshwater sediments, Spain.

*Notes: Malbranchea irregularis* is phylogenetically close to *M. californiensis* (Figure 3). Both species are morphologically well differentiated. *Malbranchea californiensis* (=*Auxarthron californiense*) produces larger ascomata (135–435 μm diam) with longer (72–375 μm), terminally uncinated and occasionally branched appendages, which are entirely smooth and arise at irregular angles around the periphery of the ascomata [74,77]. This latter feature is also observed in our new species, but the elongate appendages produced by *M. irregularis* are shorter (42.5–230.5 μm), unbranched, terminally straight, and asperulate toward the base. In addition, *M. californiensis* shows slightly bigger asci (10.6 × 7.2–8 μm vs. 6–9 × 5.5–7.5 μm in *M. irregularis*) and ascospores that are globose to ovoid (3.2–4 × 2.4–4.8 μm) and hyaline to pale yellow-brown [74,77]. The ascospores in *M. irregularis* are globose (2.5–3 μm) and hyaline. Another difference is that *M. californiensis* produces both sexual and asexual morphs, while *M. irregularis* only shows the sexual morph.

***Malbranchea sinuata*** Torres-Garcia, Gené & Cano, sp. nov. Figure 7.

*Etymology*: Referring to the sinuous fertile hyphae produced by the species.

*Type*: Spain, Catalonia, El Segrià, La Granja d’Escarp, Segre River, N 41.42754° E 0.35020°, from fluvial sediments, December 2019, D*. Torres-Garcia & J. Gené*, (**holotype** CBS H-25257; cultures ex-type FMR 18266, CBS 149938).

*Description*: *Mycelium* immersed and superficial composed of straight to sinuous, septate, branched, hyaline, thin- and smooth-walled, 1–2 μm wide hyphae. Racquet hyphae present. Asexual morph with fertile hyphae well-developed, arising as lateral branches from vegetative hyphae, right or sinuous, contorted or arcuate at apical part, 1–2 μm wide, or developing at extremes of the vegetative hyphae, in both cases forming arthroconidia randomly intercalary and terminally. *Arthroconidia* enteroarthric, 0-septate, hyaline, smooth- and thin-walled, mostly cylindrical, sometimes barrel-shaped, 1.5–3 × 1.5–2.5 μm, when mature become subglobose or irregularly shaped, secession rhexolytic. *Sexual morph* not observed.

**Figure 6 jof-09-01129-f006:**
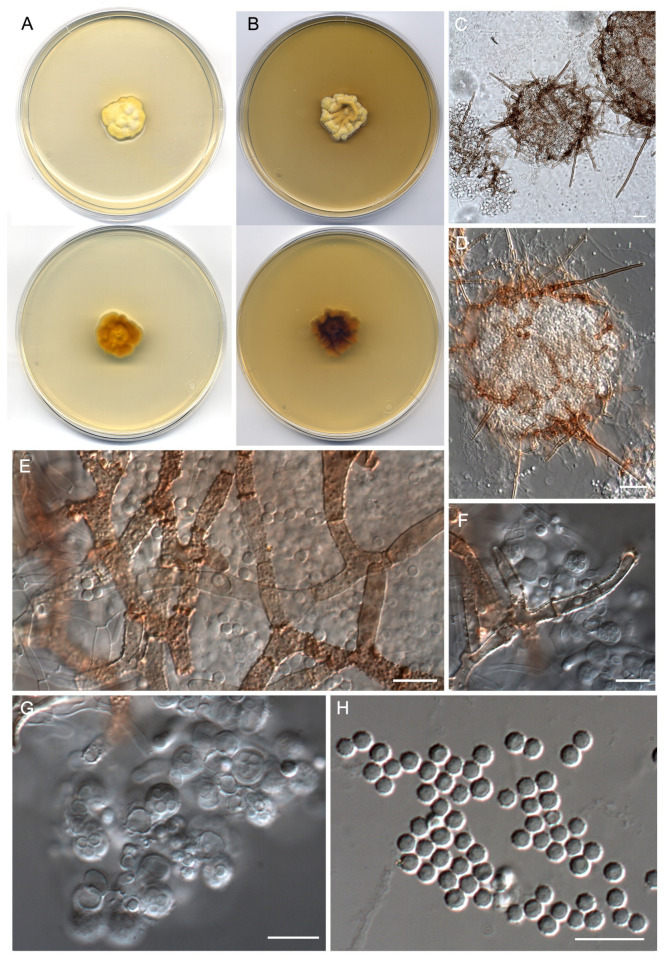
*Malbranchea irregularis* (ex-type FMR 19016). (**A**) Colonies on PDA (front and reverse) at 25 °C after 14 days. (**B**) Colonies on PYE (front and reverse) at 25 °C after 14 days. (**C**,**D**) Ascomata. (**E**) Peridial hyphae. (**F**) Peridial hyphae and asci. (**G**) Asci. (**H**) Ascospores. Scale Bars: (**C**,**D**) = 25 μm; (**E**–**H**) = 10 μm.

*Culture characteristics (14 d at 25 °C)*: Colonies on PDA reaching 54–55 mm diam., cottony, umbonate and pale yellow (4A3) to orange (5A6) at the center, flat and light orange (5A4) toward the periphery, margins regular, sporulation sparse, reverse reddish orange (7B8) at the center, orange (6A7) toward the periphery, diffusible pigment absent. On PCA, reaching 41–42 mm diam., floccose, raised, grayish orange (5B4) at the center, orange (5A6), pale orange (5A3) toward the periphery, margins regular, sporulation abundant, reverse orange (5A7) at the center, reddish yellow (4A6) at the periphery, diffusible pigment absent. On OA, reaching 40–41 mm diam., slightly raised at the center, granulose due to the production of irregular mycelial tufts, which are pale orange (5A7) to light yellow (4A4), margins irregular, sporulation abundant in the mycelial tufts, reverse brownish orange (7C8) at the center, pale orange (5A3–7) toward the periphery, diffusible pigment absent. On PYE, reaching 46–47 mm diam., cottony, slightly raised, pale yellow (1A3) to deep orange (5A8) at the center, light orange (5A4) toward the periphery, margins fimbriate, sporulation moderate, reverse reddish brown (8E6) at the center, reddish orange (7A7) toward the periphery, diffusible pigment absent.

*Cardinal temperature for growth on PDA after 14 d (mm)*: Minimum 15 °C (13–14), optimum 25 °C (49–52), maximum 37 °C (14–15).

*Habitat and geographical distribution*: Freshwater sediments, soil, and rhizosphere soil in scrublands, Spain (Figure 4, Appendix A).

*Notes: Malbranchea sinuata* is phylogenetically close to *M. albolutea*. The main difference observed between the species is that *M. sinuata* only produces the asexual morph in the media and conditions tested, while *M. albolutea* presents both sexual/asexual morphs [78]. The asexual morph of the two species produces hyaline arthroconidia by the segmentation of broader primary hyphae and of arcuate or curved lateral fertile branches, which give rise to cylindrical, barrel-shaped, subglobose or irregularly shaped conidia at maturity. However, both species differ in arthroconidial size, which is smaller in *M*. *sinuata* (1.5–3 × 1.5–2.5 μm) than in *M. albolutea* (2.5–3(–4) × (1.5–)2–5(–6.5) μm) [78], and in colony color; our new species shows colonies of yellow to orange and reddish brown shades in PYE and OA, while in *M. albolutea*, they are only of yellow shades in the same culture media previously mentioned [78]. In addition, *M. sinuata* grew at 37 °C, while *M. albolutea* was not able to grow at that temperature, except for the strain UAMH 1846 according to Sigler and Carmichael [78].

***Neoarthropsidaceae*** Torres-Garcia & Gené, fam. Nov. (Figure 2, Clade IV).

MycoBank MB850654

*Type genus*: *Neoarthropsis* Torres-Garcia, Cano & Gené.

*Included genera*: *Albidomyces*, *Apinisia*, *Arachnotheca*, *Myriodontium*, and *Neoarthropsis*.

*Description*: *Colonies* white, yellowish, or cream. *Asexual morph* arthropsis-like, chrysosporium-like, and myriodontium-like. *Sexual morph* with gymnothecial ascomata, superficial, single or aggregated, white, globose, with or without appendages, peridium composed of a network of loosely interwoven undifferentiated hyphae, septate, branched, hyaline, thin- or thick-walled, peridial appendages helical when present. *Asci* unitunicate, 8-spored, evanescent, globose, subglobose or oval, in clusters or in chains. *Ascospores* one-celled, globose, subglobose or oblate, hyaline, subhyaline, yellowish, rarely pale brown, smooth-, punctate- or punctate-reticulate wall, with or without a sheath.

*Notes*: The family *Neoarthropsidaceae* is based on a well-supported clade (80% bs/0.97 pp) in our concatenated LSU/ITS analysis (Figure 2), which correlated with the fully supported lineage, composed of practically the same taxa, shown in the multi-gene analysis previously published by Kandemir et al. [8]. However, in addition to being a highly supported lineage clearly separated from other families of the order, *Neoarthropsidaceae* is also supported because its members share morphological traits, as described above, and similar physiological features, such as having maximum temperatures for growth at 25–30 °C and the lack of, or weak, keratinolytic activity [32,36,74,75,79,80,81]. The only member reported to have strong keratinolytic activity is *My. keratinophylum* [80], which is the only species in the family described as a human opportunist by a case of sinusitis [17]. Therefore, according to the reviewed data [32,36,74,75,79,80,82,83], the species in *Neoarthropsidaceae* can be defined ecologically as saprobic, commonly inhabiting soil, rarely animal dung or plant debris, and able to survive in aquatic environments like freshwater and marine sediments. Currently, the family includes 12 species, from which those found in aquatic sediments are *Al. albicans*, *C. undulatum*, *My. keratinophylum*, *Ne. hispanica*, and *Ne*. *sexualis*.

**Figure 7 jof-09-01129-f007:**
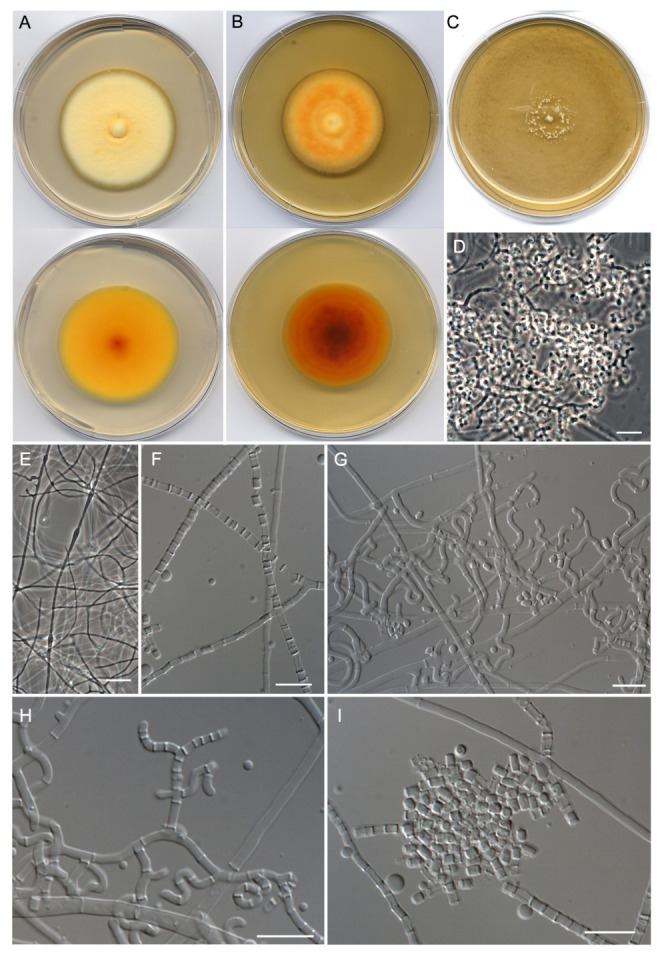
*Malbranchea sinuata* (ex-type FMR 18266). (**A**) Colonies on PDA (front and reverse) at 25 °C after 14 days. (**B**) Colonies on PYE (front and reverse) at 25 °C after 14 days. (**C**) Colony on OA (front) at 25 °C after 14 days. (**D**) Hyphae on tufts. (**E**) Racquet hyphae. (**F**) Main fertile hyphae. (**G**,**H**) Sinuous fertile hyphae. (**I**) Arthroconidia. Scale Bars: (**D**) = 25 μm; (**E–I**) = 10 μm.

***Albidomyces*** Torres-Garcia, Dania García, Cano & Gené, gen. nov.

*MycoBank* MB850655

*Etymology*: Referring to the white colony color of the type species on all agar media tested.

*Type species*: *Albidomyces albicans* (Apinis) Torres-Garcia, Dania García, Cano & Gené.

*Description*: *Colonies* white or cream, floccose or felty, composed of hyaline, septate, and branched hyphae. *Racquet hyphae* present. *Asexual morph* chrysosporium-like, with undifferentiated conidiophores. Fertile hyphae straight to flexuose, septate, branched, lateral branches arising at right angles, throughout conidia borne sessile or in short cylindrical or conical protrusions. *Conidia* terminal or lateral, one-celled, subglobose or pyriform, hyaline, smooth-walled, secession rhexolytic. *Sexual morph* gymnothecial ascomata, superficial, single or aggregated, white, globose, without peridial appendages, peridium composed of loosely interwoven undifferentiated hyphae, septate, branched, hyaline, smooth- and thin-walled. *Ascomata initials* developing in a side branch spirally twisted around ascogonium. *Asci* 8-spored, evanescent, oval, in clusters on short stalks. *Ascospores* unicellular, hyaline, and slightly punctate, globose to subglobose.

***Albidomyces albicans*** (Apinis) Torres-Garcia, Dania García, Cano & Gené, comb. nov.

MycoBank MB850658

≡ *Arachniotus albicans Apinis*, Mycological papers 96: 45, 1964; ≡ *Amauroascus albicans* (Apinis) Arx, Persoonia 6: 376, 1971; ≡ *Arachnotheca albicans* (Apinis) Arx, *The genera of fungi sporulating in pure culture*^,^ 2nd ed., J Cramer, Veduz.: 98, 1974; ≡ *Nannizziopsis albicans* (Apinis), Guarro, Cano & de Vroey, Mycotaxon 42: 195, 1991.

*Type*: UK, Lakenham, Norfolk, waterlogged alluvial pasture soils, April 1950, *Apinis*, (holotype IMI 100875, cultures ex-type CBS 151.65, ATCC 22478, BDUN 264, IHEM 4423, NRRL 5141, RV 2280).

*Description*: *Mycelium* immersed and superficial, composed of hyaline, septate, branched, smooth-walled hyphae, 1.5–3 μm wide. *Racquet hyphae* present. *Asexual morph* with fertile hyphae branched, hyaline, smooth-walled, and conidia borne sessile or in short cylindrical or conical protrusions. *Conidia* solitary, terminal, and lateral, unicellular, hyaline, smooth-walled, pyriform, 3.5–4.5(–6.5) × 4.5–7.5 μm. *Sexual morph* with gymnothecial ascomata, superficial, single or confluent, white or cream, globose, 188–435 μm diam, peridial hyphae composed of a conspicuous network of hyphae loosely interwoven, undifferentiated, septate, branched, hyaline, smooth, 2.5–3 μm wide. *Asci* 8-spored, evanescent, in clusters, oval, 8–11 × 11–17 μm. *Ascospores* unicellular, hyaline, punctate to reticulate, globose to subglobose, 3–4.5(–5) × 3–4.5 μm (adapted from Apinis [84]).

*Culture characteristics (14 d at 25 °C)*: Colonies on PDA reaching 30–31 mm diam., velvety, umbonate, radially sulcate and white (1A1) to yellowish white (1A2) at the center, flat and gray (1B1) toward the periphery, margins lobulate, sporulation absent, reverse light yellow (4A4) at the center, yellowish white (4A2) toward the periphery, diffusible pigment absent. On PCA, reaching 26–29 mm diam., velvety, concentrically sulcate and yellowish white (1A2) at the center, flat and grayish white (1B1) toward the periphery, margins lobulate, sporulation absent, reverse yellowish white (1A2) at the center, yellowish gray (2B2) toward the periphery, diffusible pigment absent. On OA, reaching 42–27 mm diam., velvety, concentrically sulcate and grayish beige (4C2) across the entire colony, immersed colony, margins entire, sporulation sparse, reverse grayish beige (4C2), diffusible pigment absent. On PYE, reaching 35–36 mm diam., velvety, umbonate, radially sulcate and white (1A1) at the center, flat and yellowish white (1A2) to white (1A1) toward the periphery, margins lobulate, sporulation absent, reverse brownish orange (5C4) at the center, grayish orange (5B5) toward the periphery, diffusible pigment absent.

*Cardinal temperature for growth on PDA after 14 d (mm)*: Minimum 15 °C (21–22), optimum 25 °C (30–31), maximum 30 °C (8–10).

*Additional specimens examined*: Spain, Aragon, Huesca, Cerler, Remáscaro Stream, N 42.58698° E 0.54993°, from river sediments, September 2018, *J.F. Cano-Lira* (FMR 17576); ibid. Catalonia, Barcelona, Bages, Balsareny, Llobregat River, N 41.87818° E 1.88900°, from river sediments, March 2021, *D. Torres-Garcia & J. Gené* (FMR 19029); ibid. Berguedà, Castellar de n’Hug, Llobregat River, N 42.24012° E 1.93742°, from river sediments, March 2021, *D. Torres-Garcia & J. Gené* (FMR 19026); ibid. (FMR 19027); ibid. (FMR 19028); ibid. Osona, Vilanova de Sau, Ter River, N 41.97084° E 2.38108°, from river sediments, September 2020, *D. Torres-Garcia & J. Gené* (FMR 18697); ibid. Girona, Llanars, Ter River, N 42.32383° E 2.32989°, from river sediments, September 2020, *D. Torres-Garcia & J. Gené* (FMR 18698); ibid. Sant Joan de les Abadesses, Ter River, N 42.24138° E 2.28830°, from river sediments, September 2020, *D. Torres-Garcia & J. Gené* (FMR 18699); ibid. Lleida, Berguedà, Gósol, Torrent de la Coma Stream, N 42.21889° E 1.65159°, from river sediments, November 2019, *J. Gené* (FMR 18029); ibid. (FMR 18030); ibid. (FMR 18031); ibid. (FMR 18033). 

*Habitat and geographical distribution*: Freshwater sediments, soil, and rhizosphere soil from different environments (forest, grassland, urban, and woodland). Czech Republic, Estonia, Germany, Latvia, Spain, and the UK (Figure 4, Appendix A).

*Notes: Arachniotus albicans* was proposed by Apinis [84] based on a group of isolates from which the selected type strain was found in alluvial soils. However, because of its inconspicuous morphological traits (i.e., white colonies, gymnothecia with undifferentiated peridial hyphae, small and globose ascospores, and an asexual morph producing terminal and lateral conidia on hyphae), its taxonomy has been confused. Arx [75] transferred the species to the genus *Amauroascus* on the basis of its ascomatal initials, spore shape and ornamentation, and its keratinolytic activity, but later transferred the species to *Arachnotheca* [85]. The last taxonomic proposal was rejected by Guarro et al. [82] because the morphological features of *Ara. albicans*, such as hyaline and sheathed ascospores or its chrysosporium-like asexual morph, did not fit with the *Arachnotheca* concept. In fact, they considered it to be a more likely species of *Nannizziopsis* [82]. The recent phylogenetic analysis of the *Onygenales* [8], which correlates with that presented here (Figure 2), shows that the generic types of the aforementioned genera are placed in other families (*Arachniotus ruber* in the *Gymnoascaceae*; *Amauroascus niger* and *Nannizziopsis vriesii* in the *Onygenaceae*), and the type of *Arachnotheca*, *Arach*. *glomerata*, although belonging to the same clade as *Ara. albicans*, is placed phylogenetically distant from the lineage representative of *Ara. albicans*. Since this later lineage, which includes the ex-type strain of Ara. albicans (CBS 151.65), was separated from the other lineages that comprised the new family *Neoarthropsidaceae*, we describe the monotypic genus *Albidomyces*. The other accepted genera included in the family can be distinguished morphologically from *Albidomyces* by the following main features: *Apinisia* by its helical appendages in the ascomata, *Arachnotheca* and *Myriodontium* by their sheathed ascospores and type of conidiogenous apparatus [8,82], and *Neoarthropsis* also by its asexual morph with enteroarthric conidia (see below). Of note is that close to the *Al. albicans* clade is the ex-type strain of *Chrysosporium sulfureum*. This species, originally described as *Isaria sulfurea*, did not form a sexual morph, and it was transferred to *Chrysosporium* by Oorschot based on the morphology of its asexual morph [79]. It has conidia growing directly on hyphae, sessile or on short cylindrical or conical protrusions. These conidial features and the ontogenesis are similar to the chrysosporium-like asexual morph present in *Al. albicans*, suggesting that it is probably another species of *Albidomyces*. However, we prefer to delay making any taxonomic changes until further investigation with more isolates of the genus.

***Neoarthropsis*** Torres-Garcia, Cano & Gené, gen. nov.

MycoBank MB847965

*Etymology*: Referring to the morphological resemblance with the genus *Arthropsis*.

*Type species*: *Neoarthropsis hispanica* (Ulfig et al.) Torres-Garcia, Cano & Gené.

*Description*: *Mycelium* septate, hyaline to subhyaline, smooth- and thin-walled, variable in width. *Racquet hyphae* usually present. *Asexual morph* arthropsis-like, consisting of fertile hyphae, single or aggregated in tufts, straight to flexuous, with lateral branches arising at right angles, straight or recurved, septating basipetally to form enteroartric conidia, often joined by distinct, narrow connectives with rhexolytic secession. *Arthroconidia* hyaline to subhyaline, straight or curved, smooth- and thin-walled, cylindrical, cubic, barrel-shaped, often broader than long, usually truncate at both ends, showing occasionally remnants of outer wall. *Sexual morph* with gymnothecial ascomata, superficial, single or confluent, growing as ascal clusters surrounded by distinct peridial hyphae, white to yellowish white, globose to subglobose, peridium composed of a conspicuous network of septate, branched, anastomosed hyphae when mature, hyaline to pale yellow, thin- and smooth-walled. *Ascomatal initials* consisting of crozier formations. *Asci* 8-spored, evanescent, borne singly or in chains, hyaline, smooth-walled, globose to subglobose or pyriform. *Ascospores* unicellular, hyaline, thick-walled, punctate (minutely echinulate), and globose.

***Neoarthropsis hispanica*** (Ulfig, Gené & Guarro) Torres-Garcia, Cano & Gené, comb. nov.

MycoBank MB850653

*Arthropsis hispanica* Ulfig, Gené & Guarro, *Mycotaxon* 54: 282, 1995.

*Type*: Spain, Catalonia, Girona, Estartit Beach, from marine sediments, September 1991, *J. Gené, K. Ulfig & J. Guarro* (holotype IMI 353700; cultures ex-type FMR 4059, CBS 352.92).

*Description*: Ulfig et al. [32] and Giraldo et al. [81].

*Additional specimens examined*: Spain, Catalonia, Barcelona, Bages, Balsareny, Llobregat River, N 41.87818° E 1.88900°, March 2021, *D. Torres-Garcia & J. Gené* (FMR 19034); ibid. (FMR 19035); ibid. Osona, Les Masies de Voltregà, Ter River, N 42.02951° E 2.25359°, from river sediments, September 2020, *D. Torres-Garcia & J. Gené* (FMR 18694).

*Habitat and geographical distribution*: Air, fluvial and marine sediments, human clinical specimens, rhizosphere soil, roots, soil from different environments (aquatic, cropland, forest, shrubland, urban, and woodland). Australia, China, Estonia, Japan, South Korea and Spain (Figure 4, Appendix A).

*Notes: Arthropsis* was established by Sigler et al. [86] and typified by *A. truncata*, to accommodate species with dark arthroconidia, joined by adjacent connectives, and developed from undifferentiated conidiogenous hyphae [87]. Later, Ulfig et al. [32] introduced *A. hispanica* based on its morphological characteristics regarding the conidial connectives and the absence of differentiated conidiophores. However, in contrast to other *Arthropsis* species, which exhibited dark pigmented mycelium and conidia, *A. hispanica* showed hyaline to subhyaline arthroconidia, fast-growing rates, and strongly tufted colonies in culture [32]. Later, based on the LSU analysis of the type strain *A. hispanica* (CBS 351.92), Giraldo et al. [87] placed the species in the order *Onygenales* (*Eurotiomycetes*), which was far away from *A. truncata* (*Sordariomycetes*). As a result, based on phylogenetic data together with the distinct morphological traits of its closely related genus *Arachnotheca* (Figure 2), we introduce the genus *Neoarthropsis* in *Onygenales* with two species, *Ne. hispanica* as the type of the genus and *Ne. sexualis* proposed and discussed below. The key morphological features to distinguish *Neoarthropsis* from *Arachnotheca* are the presence of narrow connectives in the arthroconidia of *Neoarthropsis* species (absent in the asexual morph of *Arachnotheca*) and the lack of sheath in their ascospores (presence of an irregular furrowed sheath in the ascospores of *Arachnotheca*) [74].

***Neoarthropsis sexualis*** Torres-Garcia, Cano & Gené, sp. Nov. Figure 8

MycoBank MB847966

*Etymology*: Referring to the presence of a sexual morph in the type strain.

*Type:* Spain, Catalonia, Barcelona, Bages, Balsareny, Llobregat River, N 41.87818° E 1.88900°, from fluvial sediments, March 2021, *D. Torres-Garcia & J. Gené*, (**holotype** CBS H-25259, cultures ex-type FMR 19025, CBS 149940).

*Description*: *Mycelium* immersed and superficial, composed of hyaline, septate, branched, smooth-walled, 1.5–3 μm wide hyphae. *Racquet hyphae* abundant. *Asexual morph* with undifferentiated conidiophores. *Fertile hyphae* produced mainly in tufts, hyaline, smooth- and thin-walled, main hyphae straight to flexuose, 2.5–3 μm wide, lateral branches arising at right angles, often recurved, septating basipetally to form enteroarthric conidia joined by narrow connectives. *Arthroconidia* unicellular, hyaline, thin- and smooth-walled, cylindrical or barrel-shaped, straight or curved, 2.5–7 × 1.5–3 μm. *Sexual morph* with gymnothecial ascomata, superficial, single or confluent, white to yellowish white, globose to subglobose, 365–810 μm diam, peridium composed of a conspicuous network of hyphae, septate, branched, anastomosed, hyaline to pale yellow, thick- and smooth-walled, cylindrical, 3–4 μm wide. *Asci* 8-spored, evanescent, borne singly or in chains, hyaline, smooth-walled, globose to subglobose or pyriform, 4–6 × 4–5.5 μm. *Ascospores* unicellular, hyaline, thick-walled, punctate, globose, 1.5–2.5 μm diam.

*Culture characteristics (14 d at 25 °C)*: Colonies on PDA reaching 40–41 mm diam., raised, floccose, and white (1A1) at the center, flat, cottony, and yellowish white (1A2) toward the periphery, margins slightly irregular and fimbriate, sporulation abundant, reverse pastel yellow (3A4) at the center, yellowish white (3A2) toward the periphery, diffusible pigment absent. On PCA, reaching 32–33 mm diam., slightly raised, granulose, and white (1A1) at the center, floccose to cottony and yellowish gray (2B2) toward the periphery, margins fimbriate, sporulation abundant, reverse pale yellow (4A3) at the center, yellowish white (3A2) toward the periphery, diffusible pigment absent. On OA, reaching 33–34 mm diam., granulose and white (1A1) at the center, velvety and grayish beige (4C2) toward the periphery, margins irregular, sporulation abundant, reverse grayish yellow (4B3), diffusible pigment absent. On PYE, reaching 44–45 mm diam., slightly raised at the center, floccose, white (1A1) to yellowish white (3A2), margins regular and slightly fimbriate, sporulation moderate at the center, reverse yellow (4A6) at the center, pale yellow (4A3) at the periphery, diffusible pigment absent.

*Cardinal temperature for growth on PDA after 14 d (mm*): Minimum 15 °C (16–17), optimum 25 °C (40–41), maximum 37 °C (2–3).

*Habitat and geographical distribution*: Fluvial sediment, forest, and grassland soil. Australia, Estonia, Spain, and the USA (Figure 4, Appendix A).

*Notes*: In our reconstructed phylogeny, *Ne. sexualis* is allocated in a well-supported clade with *Ne. hispanica*. The two species were recovered from aquatic environments (freshwater/marine sediments) in Spain; although *Ne. hispanica* has also been found in clinical samples in the USA, its role in disease has never been demonstrated [81]. Both species produce a practically indistinguishable asexual morph. The main difference between them is the presence of ascomata in *Ne. sexualis*, produced on OA and PCA agar, which is absent in *Ne. hispanica* under the same in vitro conditions [32,81]. In addition, *Ne. sexualis* shows more restricted growth on PDA at 25 °C after 7d (12–13 mm) and is able to grow at 37 °C, while the colonies of *Ne. hispanica* reach 17–23 mm on the same culture medium and do not grow at that temperature [81].

***Onygenales, Incertae sedis*** (Figure 2, Clade X).***Pseudoamaurascopsis*** Torres-Garcia, Dania García & Gené, gen. nov.

MycoBank MB848131

*Etymology*: Referring to the morphological resemblance with the asexual morph of the genus *Amaurascopsis*.

*Type species*: *Pseudoamaurascopsis spiralis* Torres-Garcia, Dania García & Gené.

**Figure 8 jof-09-01129-f008:**
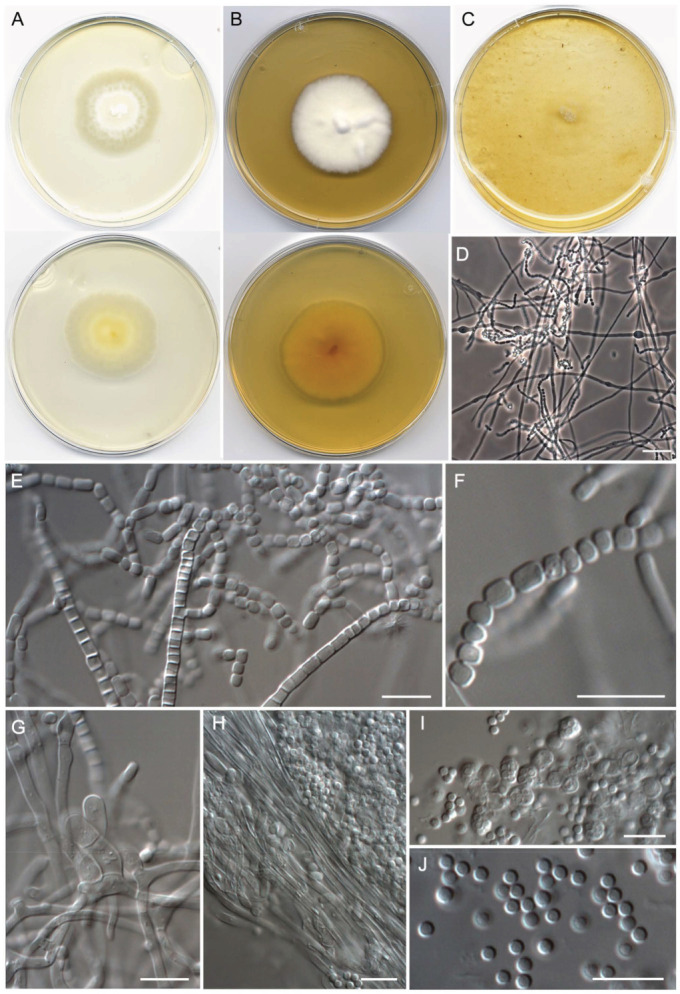
*Neoarthropsis sexualis* (ex-type FMR 19025). (**A**) Colonies on PDA (front and reverse) at 25 °C after 14 days. (**B**) Colonies on PYE (front and reverse) at 25 °C after 14 days. (**C**) Colony on OA (front) at 25 °C after 14 days. (**D**) Racquet and fertile hyphae. (**E**) Fertile hyphae. (**F**) Arthroconidia with narrow connectives. (**G**) Initial forms of asci. (**H**) Detail of a gymnothecium showing peridial hyphae and ascospores. (**I**) Asci and ascospores. (**J**) Ascospores. Scale Bars: (**D**) = 25 μm; (**E**–**J**) = 10 μm.

*Description*: *Mycelium* septate, hyaline, smooth- and thin-walled. *Spiral hyphae* present, growing laterally from vegetative hyphae. *Asexual morph* chrysosporium-like, consisting of fertile hyphae usually with intercalary and terminal thallic conidia, occasionally lateral on a short stalk, conidial secession rhexolytic. Terminal conidia arising on straight or slightly curved branches, single, hyaline to subhyaline, smooth-walled to verruculose, thick-walled, subglobose or short clavate, with a truncate end, intercalary conidia predominant, single, alternate, hyaline to subhyaline, smooth-walled to verruculose, thick-walled, cylindrical to barrel-shaped. *Sexual morph* not observed.

***Pseudoamaurascopsis spiralis*** Torres-Garcia, Dania García & Gené, sp. nov. Figure 9.

MycoBank MB848132.

*Etymology*: Referring to the ability of the species to produce spiral hyphae on vegetative hyphae.

*Type*: Spain, Catalonia, Barcelona, Bages, Balsareny, Llobregat River, N 41.87818° E 1.88900°, from fluvial sediments, March 2021, *D. Torres-Garcia & J. Gené*, (**holotype** CBS H-25260; cultures ex-type FMR 19014, CBS 149941).

*Description*: *Mycelium* partly immersed and partly superficial, composed of hyaline, septate, branched, smooth-walled, 1.5–2.5 μm wide hyphae. *Spiral vegetative hyphae* abundantly produced on OA after 14 d at 25 °C. *Asexual morph* chrysosporium-like, with terminal and intercalary thallic conidia, occasionally lateral on short stalks, terminal conidia on straight or slightly curved branches, single, hyaline to subhyaline, thick-walled, smooth-walled when young, becoming slightly echinulate, subglobose, obpyriform or short clavate, 3–5 × 2–3 μm, intercalary conidia single, alternate, hyaline to subhyaline, smooth-walled to echinulate, thick-walled, cylindrical or barrel-shaped, 7–15(–22.5) × 2–3 μm. *Sexual morph* not observed.

*Culture characteristics (14 d at 25 °C)*: Colonies on PDA reaching 23–24 mm diam., slightly raised, cottony and pinkish white (7A2) at the center and yellowish white (3B2) at the periphery, margins entire, sporulation abundant, reverse brownish orange (5C4) at the center, pale yellow (4A3) toward the periphery, diffusible pigment absent. On PCA, reaching 22–23 mm diam., slightly raised, cottony, white (1A1) to pinkish white (7A2) at the center, velvety and yellowish gray (4B2) toward the periphery, margins entire, sporulation abundant, reverse grayish orange (5B3) at the center, brownish orange (5C4) and orange-gray (5B2) toward the periphery, diffusible pigment absent. On OA, reaching 22–23 mm diam., flattened, velvety, yellowish gray (4B2), margins lobulate, sporulation absent, reverse orange-gray (5B2), diffusible pigment absent. On PYE, reaching 21–22 mm diam., raised, velvety, yellowish white (1A2) to white (1A1), margins fimbriate, sporulation sparse to moderate, reverse golden blond (5C4) at the center, pale yellow (4A4) toward the periphery, diffusible pigment absent.

*Cardinal temperature for growth on PDA after 14 d (mm*): Minimum 15 °C (5–6), optimum 25 °C (23–24), maximum 37 °C (2–3).

*Habitat and geographical distribution*: Fluvial sediments, rhizosphere soil, roots, and soil of different natural environments (cropland, grassland, and urban). Widespread distribution (Australia, Belgium, China, Denmark, Estonia, Germany, Spain, the UK, and the USA) (Figure 4, Appendix A).

*Notes*: Our phylogenetic analysis placed *Ps. spiralis* as being related to *Am. perforata* and *Po*. *hystricis*, but the three taxa were phylogenetically very distant and representatives of distinct genera. *Pseudoamaurascopsis spiralis* differs morphologically from *Am. perforata* and *Po. hystricis* mainly by the lack of a sexual morph and by the production of spiral hyphae from vegetative hyphae, which are absent at least in *Am. perforata* [88], since *Po. hystricis* shows coiled appendages but they are associated with the ascomata [89]. Despite attempts to induce the sexual morph in *Ps. spiralis*, it was never observed. The three species produce a chrysosporium-like morph with terminal and intercalary conidia, from which we only observed subtle morphological differences. The conidia of *Po. spiralis* are longer (3–11 μm) and finely echinulate, while those of *Am. perforata* are (3–)4–6.5(–7.5) μm long and roughened to tuberculate [88]. *Polytolypa hystricis* shows smooth conidia, and it is unable to grow at 37 °C [89]. *Pseudoamaurascopsis spiralis* and *Am. perforata* grow at this temperature but the latter grows faster (28–30 mm diam on PDA after 14 days) [88].

**Figure 9 jof-09-01129-f009:**
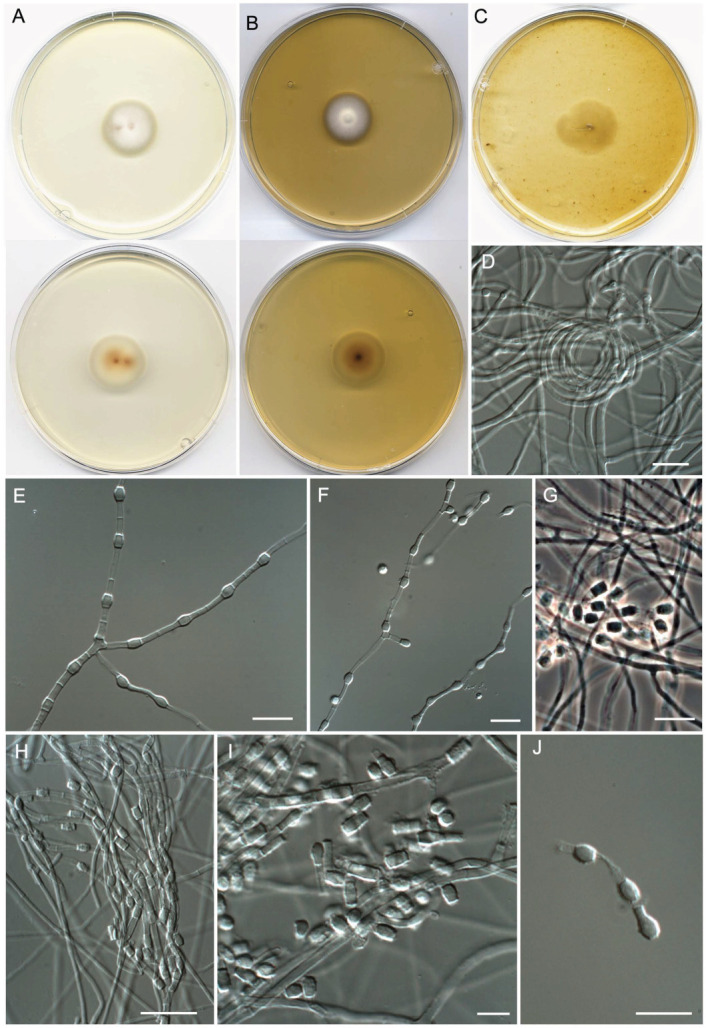
*Pseudoamaurascopsis spiralis* (ex-type FMR 19014). (**A**) Colonies on PDA (front and reverse) at 25 °C after 14 days. (**B**) Colonies on PYE (front and reverse) at 25 °C after 14 days. (**C**) Colony on OA (front) at 25 °C after 14 days. (**D**) Spiral hyphae. (**E**,**F**) Fertile hyphae with intercalary and terminal arthroconidia. (**G**−**J**) Arthroconidia. Scale Bars: (**D**–**J**) = 10 μm.

## 4. Discussion

Although the species of *Onygenales* are not considered freshwater fungi, they are able to inhabit substrates in aquatic environments such as freshwater sediments. Among onygenalean fungi found in this substrate are, for instance, the species of the thermally dimorphic nonpathogenic genus *Emmonsiellopsis*, *E. coralliformis*, and *E. terrestris*. This genus belongs to the *Ajellomycetaceae* and represents a sister lineage to the thermally dimorphic human pathogens of the family (i.e., *Blastomyces*, *Emergomyces*, *Histoplasma*, and *Paracoccidioides* (Figure 2)) [8]. In fact, the study of isolates from Spanish freshwater sediments —curiously of the same origin as the third species of the genus, *E. tuberculata*—led to the proposal of *Emmonsiellopsis* [37]. *Emmonsiellopsis tuberculata* was isolated from a sediment sample taken from an oligotrophic stream during our survey in the Aragonese Pyrenees [38]. In the present work, the study of isolates from fluvial sediments has not only allowed us to propose three novel genera (i.e., *Albidomyces*, *Neoarthorpisis*, and *Pseudoamaurascopsis*) and five new species (*M. echinulata*, *M. irregularis*, *M. sinuata*, *Ne. sexualis*, and *Ps. spiralis*), but also to delineate the new family *Neoarthropsidaceae*. According to our records, this family includes species like *Al. albicans*, *My. keratinophilum*, and *Ne. hispanica* that appear to be common in freshwater sediments. These findings and others previously reported [27,28,29,30,32,33,34,35,36,38,76] suggest that fluvial sediments are a good source of onygenalean fungi, and further investigation on their diversity is required. On that point, special attention should be paid to fluvial sediments in humid regions in the Northern Hemisphere, particularly sites in Eastern Europe, which have recently been defined as important hotspots of *Onygenales* [90].

Despite the fact that *Onygenales* are considered uncommon fungi in environmental surveys [91], using appropriate methods for their isolation can change this premise. In our survey, including PDA supplemented with cycloheximide among the culture media for fungal isolation has allowed us to isolate a set of onygenalean isolates that would otherwise have been difficult to detect, due mainly to their slow growth relative to other common environmental fungi like *Aspergillus*, *Penicillium*, or Mucorales in general. We identified 19 species of *Onygenales*, including the novel species mentioned previously (Table 1), using phenotypic traits and sequence analyses of the ITS and LSU regions. Recent molecular phylogenetic studies on the *Onygenales* have analyzed sequences of other nuclear genes like *tef*1, *tub*2, *rpb*2, or actin as the most commonly used loci to resolve the taxonomy in some groups [8,92,93,94,95,96]. However, sequences of those loci for many members of the order are not yet available for comparison. According to our results, the phylogenetic reconstruction with the barcode regions shows enough phylogenetic resolution to classify and delineate species of fresh isolates of this group of fungi. The ITS regions should be mandatory not only for identification purposes but also to assess the global distribution of the new fungi, as has been done here (see below) and in other taxonomic studies [42,73,74], including for novel species in the *Arthrodermataceae* [96].

In our study, the ITS sequence comparison of the novel taxa proposed here with the ITS environmental sequences deposited in the GlobalFungi database revealed that, with exception of the *Malbranchea* species, the other new taxa (i.e., *Al. albicans*, *Ne. hispanica*, *Ne. sexualis*, and *Ps. spiralis)* seem to be worldwide fungi that inhabit different types of soil (i.e., mainly forest, agricultural, rhizosphere soil) in various European countries and in Australia, China, South Korea, and the USA. *Neoarthropsis hispanica* was the only species with ITS environmental sequences derived from aquatic sediments different from those of the Spanish ones, namely from Japan (Appendix A). These results suggest that, even though they have been found in fluvial sediment, they are terrestrial fungi, the main origin of the *Onygenales* [9,10,11,12,13,14,15]. They probably arrive in river and stream sediments by lixiviation, where they then are able to adapt to aquatic environments. Evidence of this is that most of the abovementioned fungi shows cardinal temperatures ranging from 5 to 30 °C and are unable to growth at higher temperatures. Regarding the three new *Malbranchea* species, we only found environmental sequences related to *M. sinuata* but limited to sequences from shrubland rhizosphere soil from Spain, revealing that these *Malbranchea* species are rare fungi with a very limited distribution. However, further studies are required to confirm this approach.

Apart from the novel species, we confidently identified a set of known species, which belong to other families (Figure 1): namely, *Al. albicans* (= *Arachniotus* or *Amauroascus albicans*), *My. Keratinophilum*, and *Ne. hispanica* (= *Arthropsis hispanica*) in the *Neoarthropsidaceae*; *Ap. Crassitunicatus*, *Ap. Fulvescens*, and *Ap. Reticulisporus* in the *Onygenaceae; Ar. Curreyi* in the *Arthrodermataceae*; *E. tuberculata* in the *Ajellomycetaceae*; *Gym. Petalosporus in Gymnoascaceae*; *M. chinensis, M. ostraviensis, M. reticulata*, and *M. umbrina* in the *Malbrancheaceae*; and finally, *L. emdenii*, which is regarded as a species with an unclear taxonomic position despite being close to the *Arthrodermataceae* [8]. From those known species identified, *Ap. Fulvescens, Ap. Reticulisporus, Ar. Curreyi, Ne. hispanica, Gym. Petalosporus, L. emdenii, M. reticulata*, and *My. Keratinophilum* have already been reported in Spanish freshwater sediments in previous studies [27,29,30,32,76,80].

As we stated in the results, the most prevalent species in our survey were *Al. albicans*, *Ap. fulvescens*, and *My. keratinophilum*, which represented 55.5% of the fungi identified. Of note is that the latter two species, together with *Narasimhella marginospora* (currently *N. poonensis*), not isolated in the present survey, were found to be the most abundant species in fluvial sediments collected at the mouth of various Catalonian rivers (Besós, Ebro, Francolí, Fluvià, Llobregat, Muga, Ter, and Tordera) in the studies of Ulfig et al. [27,28,29,30]. Even in those studies, *Ap. fulvescens* and, particularly, *N. marginospora* were associated with badly polluted rivers, mainly by sewage contamination, such as Besós, Francolí, Llobregat, and Tordera. Our results are not comparable with those of Ulfig’s research, primarily because we have no data on the state of contamination of the rivers sampled in our surveys and also because sediments were collected from the upper and middle sections of the rivers rather than from the river mouth as in Ulfig’s surveys. Nevertheless, in our study, the majority of *Ap. fulvesces* and *My. keratinophilum* strains were primarily isolated from sediments of the river Segre that were collected close to rural populations and farmlands, which may have contributed to the eutrophication of the sediments. Because of the lack of experimental data, it can only be hypothesized that these two species might be bioindicators of water pollution, as stated previously by Ulfig et al. [28] regarding keratinolytic species in general. However, further studies are still needed to verify this or a similar approach. It is relevant to mention that both fungi have been described as human opportunists. Although they only cause cutaneous infections [17,97], they should be regarded as a risk to human health, particularly in river areas used for bathing purposes.

This is the first report of *Al. albicans* in fluvial sediments, although the ex-type strain (CBS 151.65) was found to be associated with wetlands, as it was isolated from pasture alluvial soils in the UK [85]. As mentioned in the taxonomy section, the classification of this species was confused; it was classified in *Amauroascus*, *Arachniotus*, *Arachnotheca*, and *Nannizziopsis* [9,75,82,85,98], and even *Kuehniella racovitzae* (currently *Apinisia racovitzae*) was considered by several authors a synonym of *Al. albicans*—at that time classified in the genus *Arachnotheca* [75,99]. As a consequence of this nomenclatural instability, data on its habitat and distribution in the literature or even in databases like the Global Biodiversity Information Facility (GBIF.org) might be confusing. After the delineation of the species in the genus *Albidomyces* (Figure 2) together with metadata on the ITS environmental sequences corresponding to *Al. albicans* deposited in the GlobalFungi database, it was revealed to be a species inhabiting mainly forests limited to European countries (Appendix A). Our data also show that *Al. albicans* is widespread in the rural and natural areas of Catalonia, where it exhibits a clear capacity to disperse and survive in both oligotrophic and eutrophic fluvial sediments. It would be interesting to verify the role of the species in aquatic environments.

The most diverse genus found in our freshwater sediment was *Malbranchea,* from which seven species were identified, including the newly proposed (Table 1). This is not so surprising since it is currently one of the most species-rich genera in the *Onygenales*, comprising nearly 30 species, many of them resistant to cycloheximide like other onygenalean fungi [8,51,78]. Ulfig et al. [30], focused on the occurrence of cycloheximide-resistant fungi from sediments collected at river mouths along the Catalonian coast, isolated several *Malbranchea* species (formerly under the *Auxarthron* genus) using Sabouraud agar supplemented with that antimicrobial agent. They were *M. arcuata*, *M. conjugata*, *M. pulchella*, *M. reticulata*, *M. umbrina,* and *M. zuffiana*, in addition to several unidentified *Malbranchea* isolates. Culture media with cycloheximide might help to investigate the diversity of *Malbranchea* in unexplored areas or substrates that would be challenging to find and isolate using other culture-dependent techniques. Taking into account that species of *Malbranchea* can usually inhabit animal dung or eutrophic environments like soil enriched by dung [8], this might lead us to think that *Malbranchea* could also be bioindicators of fecal contamination, although we lack more field surveys and laboratory tests to confirm this approach. Among the species identified here, *M. chinensis*, which was recently described from cave soil in China [83], was initially considered a rare species, but data obtained from the GlobalFungi database has revealed it is not so. ITS2 environmental sequences of *M. chinensis* were found to be associated with other types of soils, such as desert sand and rhizosphere soil, or roots from Chile and Spain; it was even detected from marine samples in China (Appendix A). Using the GlobalFungi database, we were also able to see the ubiquity in the soil of the three other species identified from Spanish fluvial samples (i.e., *M. ostraviensis*, *M. reticulata*, and *M. umbrina*), confirming the usefulness of this database for tracing the geographical distribution of fungal species over the world. It is also worth noting that *M. ostraviensis* and *M. umbrina* have been reported in cases of onychomycosis [13,77], and the latter has also been identified from various human specimens (i.e., bronchoalveolar lavage, nail, sinus, and wound), although without a demonstrated pathogenic role in disease [51].

## 5. Concluding Remarks

Although the order *Onygenales* is a well-known group of *Ascomycota*, widely investigated mainly because of its association with mammals and ability to cause superficial and systemic diseases to humans, its diversity and prevalence in aquatic environments is still poorly known. Research on its diversity in the underexplored areas of the world or poorly studied substrates, like freshwater sediments, might provide new insights and interesting specimens that allow to clarify the taxonomy of the group and also the role of these fungi in a particular habitat.

Knowing the variety and prevalence of *Onygenales* in freshwater sediments would help to determine whether they pose a risk not only to human health but also to wild animals that inhabit aquatic environments, like reptiles, in which certain onygenalean species are able to cause infections [17,95,98,100,101]. This insight would be particularly relevant in the current context of climate change [102]. It is estimated that, for instance, the rainfall patterns in the Mediterranean region will be greatly impacted by a more extreme climate [103]. Areas that are currently covered by water are likely to dry up, allowing potential pathogenic fungi like *Emmonsiellopsis* species to be exposed to the air. As a result of that and considering the ability of onygenalean fungi to adapt to extreme conditions [8], climate change may lead to an increase in human and animal infections by species previously described as nonpathogens.

## Figures and Tables

**Figure 1 jof-09-01129-f001:**
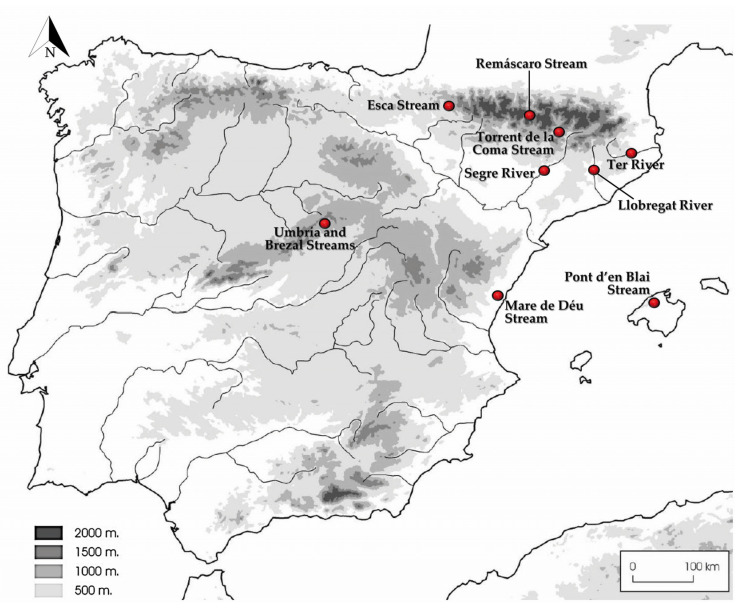
Rivers and streams sampled in Spain (red points). Altitude is represented in shades of gray. Scale bar: 100 km.

**Figure 4 jof-09-01129-f004:**
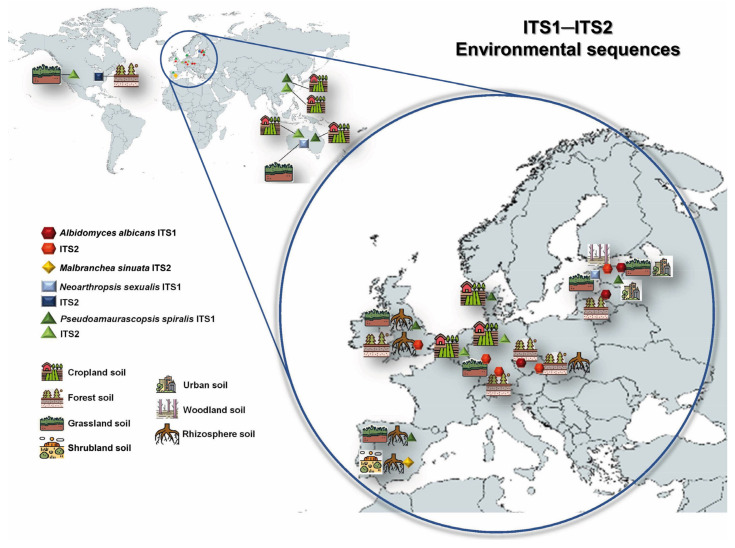
Global geographical distribution and substrate affinity of newly described taxa in *Onygenales* based on the ITS1/ITS2 environmental sequence data deposited in the GlobalFungi database (additional information in Appendix A). The European continent has been expanded for better observation of the species distribution. Polygonal shapes in different colors represent the novel taxa and their distribution according to the ITS environmental sequences detected.

**Table 1 jof-09-01129-t001:** Information and GenBank accession numbers of the isolates of the species identified in the present study.

Species	Strain Numbers ^1^	Rivers and Streams in Spain (Coordinates)	GenBank Accession Numbers			
			ITS	LSU	*tub*2	*rpb*2
***Albidomyces albicans*** (≡*Arachniotus albicans*)	FMR 17576	Remáscaro Stream, Cerler, Huesca (N 42.58698° E 0.54993°)	ON720226	ON720765	-	-
	FMR 18029	Torrent de la Coma Stream, Lleida (N 42.21889° E 1.65159°)	ON721307	ON720780	-	-
	FMR 18030	Torrent de la Coma Stream, Lleida (N 42.21889° E 1.65159°)	ON720223	ON720762	-	-
	FMR 18031	Torrent de la Coma Stream, Lleida (N 42.21889° E 1.65159°)	ON720231	ON720770	-	-
	FMR 18033	Torrent de la Coma Stream, Lleida (N 42.21889° E 1.65159°)	ON720228	ON720767	-	-
	FMR 18697	Ter River, Vilanova de Sau, Barcelona (N 41.97084° E 2.38108°)	OP373730	OP373734	-	-
	FMR 18698	Ter River, Llanars, Girona (N 42.32383° E 2.32989°)	ON720224	ON720763	-	-
	FMR 18699	Ter River, Sant Joan de les Abadesses, Girona (N 42.24138° E 2.28830°)	OP373731	OP373735	-	-
	FMR 19026	Llobregat River, Castellar de n’Hug, Barcelona (N 42.24012° E 1.93742°)	ON720229	ON720768	-	-
	FMR 19027	Llobregat River, Castellar de n’Hug, Barcelona (N 42.24012° E 1.93742°)	ON720225	ON720764	-	-
	FMR 19028	Llobregat River, Castellar de n’Hug, Barcelona (N 42.24012° E 1.93742°)	ON720227	ON720766	-	-
	FMR 19029	Llobregat River, Balsareny, Barcelona (N 41.87818° E 1.88900°)	ON720230	ON720769	-	-
*Aphanoascus crassitunicatus*	FMR 18700	Ter River, Vilanova de Sau, Barcelona (N 41.97084° E 2.38108°)	ON720203	ON720742	-	-
	FMR 18701	Ter River, Roda de Ter, Barcelona (N 41.96974° E 2.31211°)	ON720204	ON720743	-	-
	FMR 18702	Ter River, Ripoll, Girona (N 42.17258° E 2.19372°)	ON721311	ON720784	-	-
*Aphanoascus fulvescens*	FMR 18235	Segre River, Aitona, Lleida (N 41.97084° E 2.38108°)	ON720208	ON720747	-	-
	FMR 18236	Segre River, La Granja d’Escarp, Lleida (N 41.42754° E 0.35020°)	ON720210	ON720749	-	-
	FMR 18241	Segre River, La Granja d’Escarp, Lleida (N 41.42754° E 0.35020°)	ON721308	ON720781	-	-
	FMR 18242	Segre River, La Granja d’Escarp, Lleida (N 41.42754° E 0.35020°)	ON720211	ON720750	-	-
	FMR 18243	Segre River, Camarassa, Lleida (N 41.89834° E 0.88110°)	ON942224	ON942226	-	-
	FMR 18704	Ter River, Les Masies de Voltregà, Barcelona (N 42.02951° E 2.25359°)	ON720207	ON720746	-	-
	FMR 19018	Mare de Déu Stream, Borriana, Castellón (N 39.87727° W 0.05374°)	ON720212	ON720751	-	-
	FMR 19019	Mare de Déu Stream, Borriana, Castellón (N 39.87727° W 0.05374°)	ON720209	ON720748	-	-
	FMR 19022	Mare de Déu Stream, Borriana, Castellón (N 39.87727° W 0.05374°)	ON720206	ON720745	-	-
	FMR 19020	Llobregat River, Cercs, Barcelona (N 42.10512° E 1.88044°)	ON721312	ON720785	-	-
	FMR 19021	Llobregat River, Cercs, Barcelona (N 42.10512° E 1.88044°)	ON720205	ON720744	-	-
*Aphanoascus reticulisporus*	FMR 18004	Esca Stream, Navarra (N 42.81466° W 0.95221°)	ON720200	ON720739	-	-
	FMR 19012	Llobregat River, Cercs, Barcelona (N 42.10512° E 1.88044°)	ON720201	ON720740	-	-
	FMR 19013	Llobregat River, Cercs, Barcelona (N 42.10512° E 1.88044°)	ON720202	ON720741	-	-
	FMR 19033	Llobregat River, La Pobla de Lillet, Barcelona (N 42.24294° E 1.96998°)	OP373732	OP373736	-	-
*Arthroderma curreyi*	FMR 19038	Llobregat River, Balsareny, Barcelona (N 41.87818° E 1.88900°)	ON720238	ON720777	-	-
*Emmonsiellopsis tuberculata*	FMR 17582^T^	Remáscaro Stream, Cerler, Huesca (N 42.58698° E 0.54993°)	LR598892	LR598891	-	-
*Gymnoascoideus petalosporus*	FMR 19036	Llobregat River, Cercs, Barcelona (N 42.10512° E 1.88044°)	ON720236	ON720775	-	-
*Leucothecium emdenii*	FMR 18703	Ter River, Sant Joan de les Abadesses, Girona (N 42.24138° E 2.28830°)	ON720239	ON720778	-	-
*Malbranchea chinensis*	FMR 18267	Segre River, Camarassa, Lleida (N 41.89834° E 0.88110°)	ON720190	ON720729	OP425706	OP425715
***Malbranchea echinulata*** (=*Malbranchea* sp. II)	FMR 17906 ^T^ = CBS 149936	Umbría Stream, Rascafría, Madrid (N 40.86011° W 3.90849°)	ON720198	ON720737	OP425705	-
***Malbranchea irregularis*** (=*Malbranchea* sp. III)	FMR 19016 ^T^ = CBS 149937	Llobregat River, Castellar de n’Hug, Barcelona (N 42.24012° E 1.93742°)	ON720191	ON720730	OP425710	OP425719
	FMR 19017	Llobregat River, Castellar de n’Hug, Barcelona (N 42.24012° E 1.93742°)	ON720192	ON720731	OP425713	OP425722
	FMR 19030	Llobregat River, Castellar de n’Hug, Barcelona (N 42.24012° E 1.93742°)	ON720193	ON720732	OP425712	OP425721
	FMR 19015	Llobregat River, Balsareny, Barcelona (N 41.87818° E 1.88900°)	ON720194	ON720733	OP425711	OP425720
*Malbranchea ostraviensis*	FMR 18693	Ter River, Roda de Ter, Barcelona (N 41.96974° E 2.31211°)	ON720199	ON720738	OP425707	OP425716
*Malbranchea reticulata*	FMR 18696	Ter River, Roda de Ter, Barcelona (N 41.96974° E 2.31211°)	ON721310	ON720783	-	-
***Malbranchea sinuata*** (=*Malbranchea* sp. I)	FMR 18266 ^T^ = CBS 149938	Segre River, La Granja d’Escarp, Lleida (N 41.42754° E 0.35020°)	ON720195	ON720734	OP425704	OP425714
*Malbranchea umbrina*	FMR 17899	Ter River, Sant Joan de les Abadesses, Girona (N 42.24138° E 2.28830°)	ON721306	ON720779	-	-
	FMR 18695	Ter River, Sant Joan de les Abadesses, Girona (N 42.24138° E 2.28830°)	ON720196	ON720735	OP425709	OP425718
	FMR 18766	Brezal Stream, Rascafría, Madrid (N 40.86011° W 3.90849°)	ON720197	ON720736	OP425708	OP425717
*Myriodontium keratinophilum*	FMR 17624	Pont d’en Blai Stream, Mallorca, Balears Islands (N 39.76907° E 2.88418°)	ON720220	ON720759	-	-
	FMR 18244	Segre River, Aitona, Lleida (N 41.48444° E 0.46446°)	ON720222	ON720761	-	-
	FMR 18247	Segre River, Aitona, Lleida (N 41.48444° E 0.46446°)	ON721309	ON720782	-	-
	FMR 18248	Segre River, Aitona, Lleida (N 41.48444° E 0.46446°)	ON720215	ON720754	-	-
	FMR 18245	Segre River, La Granja d’Escarp, Lleida (N 41.42754° E 0.35020°)	ON720213	ON720752	-	-
	FMR 18246	Segre River, La Granja d’Escarp, Lleida (N 41.42754° E 0.35020°)	ON720221	ON720760	-	-
	FMR 18251	Segre River, La Granja d’Escarp, Lleida (N 41.42754° E 0.35020°)	ON720217	ON720756	-	-
	FMR 18258	Segre River, La Granja d’Escarp, Lleida (N 41.42754° E 0.35020°)	ON720219	ON720758	-	-
	FMR 18249	Segre River, Camarassa, Lleida (N 41.89834° E 0.88110°)	ON720216	ON720755	-	-
	FMR 18250	Segre River, Camarassa, Lleida (N 41.89834° E 0.88110°)	ON720218	ON720757	-	-
	FMR 18257	Segre River, Camarassa, Lleida (N 41.89834° E 0.88110°)	ON720214	ON720753	-	-
	FMR 19031	Llobregat River, Guardiola del Berguedà, Barcelona (N 42.24012°E 1.93742°)	OP070725	OP077087	-	-
***Neoarthropsis hispanica*** (≡*Arthropsis hispanica*)	FMR 18694	Ter River, Les Masies de Voltregà, Barcelona (N 42.02951° E 2.25359°)	ON720234	ON720773	-	-
	FMR 19034	Llobregat River, Balsareny, Barcelona (N 41.87818° E 1.88900°)	ON720232	ON720771	-	-
	FMR 19035	Llobregat River, Balsareny, Barcelona (N 41.87818° E 1.88900°)	ON720233	ON720772	-	-
** *Neoarthropsis sexualis* **	FMR 19025^T^ = CBS 149940	Llobregat River, Balsareny, Barcelona (N 41.87818° E 1.88900°)	ON720235	ON720774	-	-
** *Pseudoamaurascopsis spiralis* **	FMR 19014^T^ = CBS 149941	Llobregat River, Balsareny, Barcelona (N 41.87818° E 1.88900°)	ON720237	ON720776		

^1^ CBS: culture collection of the Westerdijk Fungal Biodiversity Institute, Utrecht, The Netherlands; FMR: Facultat de Medicina i Ciències de la Salut, Reus, Spain; the taxonomic novelties proposed are in bold; ^T^ indicates ex-type strains.

## Data Availability

The data presented in this study are available on request from the corresponding author.

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
