# Peer review of "Insights into Some Onygenalean Fungi from Freshwater Sediments in Spain and Description of Novel Taxa"

_jof, 2023, doi:10.3390/jof9121129_

Round 1

Reviewer 1 Report

Comments and Suggestions for Authors

It is a good paper describing new taxa supported by morphological and molecular data. 

As stated in the pdf file with comments some sections are mixed. For instance, the last two sentences of the introduction correspond to methodology. While in the phylogenetic results, there are some comments about morphology that should be placed under the notes of each taxa. 

I suggest adding a map of the sampling points instead of the full information in table 1 (this information is also stablished in the specimens examined).

Regarding to the description of the new taxa. I would like to ask the authors to re-consider:

The use of “Released by rhexolytic secession” and use only “rhexolytic secession” (release is implicit in secession)

Cardinal temperatures: the "no growth" information it is not necessary (since authors already given the minimum and maximum temperatures).

Additional specimens examined. If different strains were collected in the same place, it is not necessary to write again all the information (see comment in the pdf)

Habitat and geographical distribution. I add suggestions in the format to improve the presentation of this information. 

Please find additional comments/suggestions on the pdf document.

Author Response

Responses to Reviewer 1

We thank you very much for the comments and corrections in the pdf file. We hope changes introduced improve the manuscript.

- As stated in the pdf file with comments some sections are mixed. For instance, the last two sentences of the introduction correspond to methodology. While in the phylogenetic results, there are some comments about morphology that should be placed under the notes of each taxa. 

Response: All this has been reorganized as stated the reviewer. Nevertheless, we have kept at the end of the introduction the aim of assessing the global distribution and habitat affiliation of the novel fungi described in the paper because it is necessary to state here part of the information that is given in the results and also discussed.

- I suggest adding a map of the sampling points instead of the full information in table 1 (this information is also stablished in the specimens examined).

Response: We added a map indicating the rivers and streams sampled. However, because Table 1 includes not only the novel taxa proposed, but also all the strains identified in our study that are not included in any other section of the text, we have maintained metadata for all the strains.

- Regarding to the description of the new taxa. I would like to ask the authors to re-consider: The use of “Released by rhexolytic secession” and use only “rhexolytic secession” (release is implicit in secession)

Response: This has been modified accordingly, as have most of the corrections given in the pdf file for the descriptions, with the exception of a very few that, according to our experience, are commonly used in fungal descriptions.

- Cardinal temperatures: the "no growth" information it is not necessary (since authors already given the minimum and maximum temperatures).

Response: This has been deleted in the entire Taxonomy section.

- Additional specimens examined. If different strains were collected in the same place, it is not necessary to write again all the information (see comment in the pdf)

Response: This has been modified accordingly.

- Habitat and geographical distribution. I add suggestions in the format to improve the presentation of this information. 

Response: Suggestions of the reviewer has been included.

- Please find additional comments/suggestions on the pdf document.

Response: We have read and carefully included in the new version practically all the suggestions and corrections given by the reviewer.

Comments or questions of this reviewer included in the pdf file are answered below:

-Why do you only mention only PDA here? In material and method you refer to also DRBC

Response: We mentioned this medium in the introduction because it has been used previously by other authors to isolate onygenalean fungi, as stated previously in this section.

-Regarding UAMH 1846: Why you didn’t include this strain in your analysis?

Response: We did not include this strains in our analyses since its sequences were not available for comparison.

- In previous description you inlude racquet hyphae as part of the asexual morph. Please correct accordingly.

Response: The sort of hyphae (racquet or spiral hyphae) is described before the morphological traits of the asexual morph in all descriptions.

-Abbreviations in notes of Albidomyces albicans.

Response: Mainly to avoid confusion for the reader, we think that generic abbreviations are not necessary at this point since all species mentioned here are not mentioned in other parts of the text.

- Regarding Notes on Neoarthorpsis sexualis: this is not mention in habitat.

Response: in Notes of this species “clinical specimens are referred to Ne. hispanica not to Ne. sexualis. In Habitat of Ne hispanica this is mentioned.

-Comment in leyend of Fig. 7: please be more specific. Which kind of tissue is that?

Response: since it is a gymnothecium and the peridium is composed of a network of hyphae, we cannot mention a particular tissue for the peridium. Nevertheless, we have modified this part of the legend with “Detail of a gymnothecium, showing peridial hyphae and ascospores”.

-Spiral or coiled hyphae

Response: the term spiral is also common for this hyphal form.

-In other description you used the term vegetative hyphae. I there a difference among this two terms? if not please be consistent in the use over the other descriptions

Response: Descriptions of the microscopic features mostly begin with mycelial features. We refer to vegetative hyphae as a supporting structure from which arise fertile hyphae. These terms are commonly used in descriptions of microscopic fungi.

Reviewer 2 Report

Comments and Suggestions for Authors

This manuscript is well-organized and worth publishing. The authors have conducted ITS-LSU phylogenetic analyses of Onygenales, one new family (Neoarthropsidaceae), and three new genera (Albidomyces, Neoarthropsis, and Pseudoamaurascopsis) were introduced to this order. Moreover, three new species of Malbranchea were also provided based on multigene phylogeny (ITS, LSU, RPB2, and TUB2) and morphology. The manuscript is based on solid methods, and the results are convincing and justified. Publication is recommended after minor revision.

Author Response

Responses to Reviewer 2 (Minor revision)

This manuscript is well-organized and worth publishing. The authors have conducted ITS-LSU phylogenetic analyses of Onygenales, one new family (Neoarthropsidaceae), and three new genera (Albidomyces, Neoarthropsis, and Pseudoamaurascopsis) were introduced to this order. Moreover, three new species of Malbranchea were also provided based on multigene phylogeny (ITS, LSU, RPB2, and TUB2) and morphology. The manuscript is based on solid methods, and the results are convincing and justified. Publication is recommended after minor revision.

We appreciate the positive comments provided by the reviewer about our work. However, we did not find any attached file with recommendations.

Reviewer 3 Report

Comments and Suggestions for Authors

Review for Journal of Fungi: Onygenales

This is an excellent account of members of the Onygenales isolated from rivers in Spain. Two new genera and several new species were discovered and are described and illustrated here. The descriptions and illustrations are excellent. The basis for the identifications and new taxa are supported by molecular phylogeny of all isolates. The results are augmented by ITS sequences from GlobalFungi to round out the distribution of the isolated and newly described taxa. A few misspellings were found and must be corrected. The English is good, although some of the stylistic features could be improved especially use of extra words such as however and therefore where they are not needed. Proper names should be listed in alphabetical order unless some other order is required. Random is not a kind of order. The author abbreviations for new taxa must be in accordance with the International Plant Name Index and generally are correct except for Cano, which should be Cano-Lira.  Also, usually the author list for a taxon uses an ampersand rather than the word “and”.

Below are editorial suggestions as well as the misspellings.

Abstract: 

Ln 12, delete However

Lns. 14-15, proper names in alphabetical order

Ln 16, misspelled Marbranchea, should be Malbranchea

Ln 17, delete “a” before “new genera”, change to The new genera…

Ln 20, a sexual morph

Keywords: Use semicolon between all words/phrases.

Introduction

Ln 35, delete “The”

Ln 43-44, divide into two sentences

Ln 49-50, delete “It is also widely known that…”

Ln 55, delete “There are…” and “that” so it reads “Several studies have…:

Ln 56-57, delete “It is interesting to draw attention to the…” Thus, “Studies by…explore the diversity…”

Ln 59, culture-dependent techniques

Ln 67, 69, 87, 100, 101, and throughout, proper names should be listed alphabetically

Ln 70, delete “might”

Ln 88, there is no such word as “polyphylia”. Change to ‘’polyphyletic nature”

Ln 100, redundant, “formally existing; delete “already”, thus “to the seven existing families”

Ln 102-103, delete this sentence

Ln 108, delete “therefore”

Materials and Methods

Table 1, it would be useful to include the CBS numbers

Also, what does it mean if the coordinates are missing? Are they the same as the ones above? Or are they unknown. There should be something in these spaces.

Albidomyces albicans triple equals Arachniotus albicans

Ln 163-164, change “were for assessing” to “were used to assess”

Ln 230, delete extra space and period between sentences

Results

Ln 266, delete “the”

Ln 274, use an en dash

Ln 295-6, put in alphabetical order

Ln 302, change “matched” to “was part of” or “formed”

Ln 377-8, put in alphabetical order

Ln 390, change “allow us” to “provide evidence for proposing”

Ln 411, Gdeus may be mistaken for the generic name, not an abbreviation, Either write out the name or make the abbreviation with a period so that the reader knows it is an abbreviation

Ln 418, delete “precisely”

Ln 427, this phrase doesn’t make sense especially the phrase “need for taxonomic adjustment”

Ln 428, change “prevent” to “preclude”

Ln 461-62, put in alphabetical order

Taxonomy

Ln 522 and throughout: after Etymology it is not necessary to include the word Name as the Etymology  refers to the name, thus the word Name could be deleted after Etymology everywhere.

Ln 522, also, use the term wall ornamentation

Ln 534, echinulate is misspelled as equinulate a number of places. Check throughout.

Ln 537, 538, 550, 587, 586, 639, 640, and elsewhere, delete “the” in formal descriptions throughout the descriptions.

Ln 540, in the description, think-walled is use. I can’t tell if this should be thick or thin!

Ln 542, 593, 646, 750 and elsewhere: put a space between 14 and d as done further down in the description

Fig. 4, the peridial hyphae do not look echinulate as mentioned in the description

Ln 565, why is there a hyphen at the end of the line?

Ln 654, place a comma before which

Ln 701, change shear to sheer or delete this word

Ln 703, keratinolytic is misspelled in several places

Ln 704, delete “precisely”

Ln 709, delete “nearly of” to “includes”, delete “at the moment”. Everything is at the moment.

Ln 719, change “may be” to something such as often, occasionally or usually.

Ln 721, delete “throughout”

Ln 726, change “consisting” to “developing”

Ln 727, delete “the”

Ln 737, put “Apinis” in italics because he was the collector as done in other specimen citations.

Ln 740, change “in” to “of”

Ln 745, change “by” to “composed of”

Ln 783, Apart is one word

Ln 791, delete “particularly”

Ln 792, misspelled “gymnothecia

Ln 794, place a comma after the close parenthesis

Ln 796, misspelled “keratinolytic

Ln 800, delete “Nevertheless”

Ln 818, change “supporting” to “suggesting”

Ln 819, change “preferred” to “prefer”

Ln 832, delete “the”

Ln 837, borne singly

Ln 858, change to aquatic

Ln 866, change to mycelium

Ln 897, borne singly

Ln 935, consisting of

Ln 937, delete “the”

Ln 959, 961, misspelled “echinulate”

Ln 979, Apart is one word

Ln 985, First sentence of notes—what does this mean? How can you say that these taxa were close but distant?

Discussion

Ln 1000, change “like” to “such as”

Ln 1029, comma after isolation?

Ln 1031, delete “then”

Ln 1033, delete “Even”

Ln 1049, delete “Meanwhile”

Ln 1054, delete “confidentially”. Do you mean confidently?

Ln 1055, comma before which

Ln 1084, comma before although. Break this sentence into two sentences.

Ln 1089, what is conflictive? Perhaps you mean varied or confused. The latter has been considered. Arachnotheca is the correct spelling.

Ln 1094, delete “However”

Ln 1103, delete “However” Insert “most”

Ln 1108, change “through” to “using”

Ln 1115, Malbranchia is misspelled

Ln 1119, 1123, GlobalFungi should have a capital F

Ln 1122, change “through” to “using”

Ln 1124, put in alphabetical order

Ln 1129, “should be”

Ln 1131, change “Despite that” to “Although”

Ln 1136, Delete “help to advance knowledge” and change “but” to “as well as develop”

Ln 1144, delete “So…”

Comments on the Quality of English Language

/

Author Response

Responses to Reviewer 3.

Authors thank reviewer 3 for the comments and corrections to improve the content of the ms.

- A few misspellings were found and must be corrected. The English is good, although some of the stylistic features could be improved especially use of extra words such as however and therefore where they are not needed.

Response: All misspellings and stylistic features have been checked and corrected.

- Proper names should be listed in alphabetical order unless some other order is required. Random is not a kind of order.

Response: All names have been listed in alphabetical order.

- The author abbreviations for new taxa must be in accordance with the International Plant Name Index and generally are correct except for Cano, which should be Cano-Lira.

Response: We checked the author abbreviation list and saw that for JF Cano (=JF Cano-Lira), there are two possibilities. Therefore, since the name “Cano” was the most used form a long time ago by the author, this is the name that the author prefers to keep in the authorities for the novel taxa proposed in the present study.

- Also, usually the author list for a taxon uses an ampersand rather than the word “and”.

Response: This has been changed accordingly.

- Below are editorial suggestions as well as the misspellings.

- In Abstract: 

Ln 12, delete However. Done

Lns. 14-15, proper names in alphabetical order. Done

Ln 16, misspelled Marbranchea, should be Malbranchea. Done

Ln 17, delete “a” before “new genera”, change to The new genera… Done

Ln 20, a sexual morph. Done

Keywords: Use semicolon between all words/phrases. Done

- In Introduction

Ln 35, delete “The”. Done

Ln 43-44, divide into two sentences. Done

Ln 49-50, delete “It is also widely known that…”. Done

Ln 55, delete “There are…” and “that” so it reads “Several studies have…: Done

Ln 56-57, delete “It is interesting to draw attention to the…” Thus, “Studies by…explore the diversity…”. Done

Ln 59, culture-dependent techniques. Done

Ln 67, 69, 87, 100, 101, and throughout, proper names should be listed alphabetically. 

Response. We followed the recommendation. Due to these changes the order of some reference has also changed. Therefore, the section References has been modified accordingly.

Ln 70, delete “might”. Done

Ln 88, there is no such word as “polyphylia”. Change to ‘’polyphyletic nature”. Done

Ln 100, redundant, “formally existing; delete “already”, thus “to the seven existing families” . Done

Ln 102-103, delete this sentence. Done

Ln 108, delete “therefore”. Done

- In Materials and Methods

- Table 1, it would be useful to include the CBS numbers. Also, what does it mean if the coordinates are missing? Are they the same as the ones above? Or are they unknown. There should be something in these spaces.

Response: CBS numbers has been included, as well as the missing information for the rest of strains.

- Albidomyces albicans triple equals Arachniotus albicans.  Done

Ln 163-164, change “were for assessing” to “were used to assess” Done

Ln 230, delete extra space and period between sentences. Done

- In Results

Ln 266, delete “the”. Done

Ln 274, use an en dash. Changed

Ln 295-6, put in alphabetical order. Done

Ln 302, change “matched” to “was part of” or “formed”. Done

Ln 377-8, put in alphabetical order. Done

Ln 390, change “allow us” to “provide evidence for proposing”. Done

Ln 411, Gdeus may be mistaken for the generic name, not an abbreviation, Either write out the name or make the abbreviation with a period so that the reader knows it is an abbreviation. Done

Ln 418, delete “precisely”. Done

Ln 427, this phrase doesn’t make sense especially the phrase “need for taxonomic adjustment” Done

Ln 428, change “prevent” to “preclude”. Done

Ln 461-62, put in alphabetical order. Done

- In Taxonomy

Ln 522 and throughout: after Etymology it is not necessary to include the word Name as the Etymology  refers to the name, thus the word Name could be deleted after Etymology everywhere.

Response: It has been modified accordingly

Ln 522, also, use the term wall ornamentation. Done

Ln 534, echinulate is misspelled as equinulate a number of places. Check throughout. Done

Ln 537, 538, 550, 587, 586, 639, 640, and elsewhere, delete “the” in formal descriptions throughout the descriptions. Done

Ln 540, in the description, think-walled is use. I can’t tell if this should be thick or thin!. Done

Ln 542, 593, 646, 750 and elsewhere: put a space between 14 and d as done further down in the description.  Done

Fig. 4, the peridial hyphae do not look echinulate as mentioned in the description

Response: We checked this feature and changed it by “finely echinulate”.

Ln 565, why is there a hyphen at the end of the line? 

Response: It was deleted

Ln 654, place a comma before which. Done

Ln 701, change shear to sheer or delete this word. Done

Ln 703, keratinolytic is misspelled in several places. Done

Ln 704, delete “precisely” . Done

Ln 709, delete “nearly of” to “includes”, delete “at the moment”. Everything is at the moment. Done

Ln 719, change “may be” to something such as often, occasionally or usually. Done

Ln 721, delete “throughout” . Done

Ln 726, change “consisting” to “developing”. Done

Ln 727, delete “the”. Done

Ln 737, put “Apinis” in italics because he was the collector as done in other specimen citations. Done

Ln 740, change “in” to “of”

Response: There is not the word “in” in Ln 740 of the first version of the ms.

Ln 745, change “by” to “composed of”. Done

Ln 783, Apart is one word. Done

Ln 791, delete “particularly”. Done

Ln 792, misspelled “gymnothecia” . Done

Ln 794, place a comma after the close parenthesis . Done

Ln 796, misspelled “keratinolytic” . Done

Ln 800, delete “Nevertheless” . Done

Ln 818, change “supporting” to “suggesting”. Done

Ln 819, change “preferred” to “prefer” . Done

Ln 832, delete “the” . Done

Ln 837, borne singly. Done

Ln 858, change to aquatic. Done

Ln 866, change to mycelium. Done

Ln 897, borne singly. Done

Ln 935, consisting of. Done

Ln 937, delete “the”. Done

Ln 959, 961, misspelled “echinulate”. Done

Ln 979, Apart is one word. Done

Ln 985, First sentence of notes—what does this mean? How can you say that these taxa were close but distant? Changed

- In Discussion

Ln 1000, change “like” to “such as” Done

Ln 1029, comma after isolation? 

Response: Sorry, but there is not the word “isolation” in Ln 1029 of the first version of the ms.

Ln 1031, delete “then” . Done

Ln 1033, delete “Even” . Done

Ln 1049, delete “Meanwhile”. Done

Ln 1054, delete “confidentially”. Do you mean confidently? Done

Ln 1055, comma before which Done

Ln 1084, comma before although. Break this sentence into two sentences. Done

Ln 1089, what is conflictive? Perhaps you mean varied or confused. The latter has been considered. Arachnotheca is the correct spelling. Done

Ln 1094, delete “However” Done

Ln 1103, delete “However” Insert “most”. Done

Ln 1108, change “through” to “using”. Done

Ln 1115, Malbranchia is misspelled. Corrected

Ln 1119, 1123, GlobalFungi should have a capital F. Changed

Ln 1122, change “through” to “using” . Done

Ln 1124, put in alphabetical order. Done

Ln 1129, “should be”. 

Response: We do not understand properly this correction.

Ln 1131, change “Despite that” to “Although”. Done

Ln 1136, Delete “help to advance knowledge” and change “but” to “as well as develop”

Response: Following the reviewer corrections, we have simplified this sentence.

Ln 1144, delete “So…” . Done